# Three-dimensional synaptic organization of the human hippocampal CA1 field

**Marta Montero-Crespo[1,2], Marta Dominguez-Alvaro[2], Patricia Rondon-Carrillo[2], Lidia Alonso-Nanclares[1,2,3], Javier DeFelipe[1,2,3], Lidia Blazquez-Llorca[2,4]***

[1]Instituto Cajal, Consejo Superior de Investigaciones Científicas (CSIC), Madrid, Spain; [2]Laboratorio Cajal de Circuitos Corticales, Centro de Tecnología Biomédica, Universidad Politécnica de Madrid, Madrid, Spain; [3]Centro de Investigación Biomédica en Red sobre Enfermedades Neurodegenerativas (CIBERNED), ISCIII, Madrid, Spain; [4]Departamento de Psicobiología, Facultad de Psicología, Universidad Nacional de Educación a Distancia (UNED), Madrid, Spain

**Abstract** The hippocampal CA1 field integrates a wide variety of subcortical and cortical inputs, but its synaptic organization in humans is still unknown due to the difficulties involved studying the human brain via electron microscope techniques. However, we have shown that the 3D reconstruction method using Focused Ion Beam/Scanning Electron Microscopy (FIB/SEM) can be applied to study in detail the synaptic organization of the human brain obtained from autopsies, yielding excellent results. Using this technology, 24,752 synapses were fully reconstructed in CA1, revealing that most of them were excitatory, targeting dendritic spines and displaying a macular shape, regardless of the layer examined. However, remarkable differences were observed between layers. These data constitute the first extensive description of the synaptic organization of the neuropil of the human CA1 region.

*For correspondence:
lblazquez@psi.uned.es

**Competing interests:** The authors declare that no competing interests exist.

## Introduction

The hippocampus plays a crucial role in spatial orientation, learning and memory, and many pathological conditions (e.g., epilepsy and Alzheimer's disease) are closely associated with synaptic alterations in the hippocampus (*Amaral and Lavenex, 2007*). As has been previously discussed, one of the first steps towards understanding the way in which neuronal circuits contribute to the functional organization of the brain involves defining the brain's detailed structural design and mapping its connection matrix (*Swanson and Bota, 2010*). The connectivity of the brain can be examined at three major levels of resolution (*DeFelipe, 2010*): (i) macroscopically, focusing on major tract connectivity; (ii) at an intermediate resolution, using light microscopy techniques that allow putative synaptic contacts to be mapped; and (iii) at the ultrastructural level, using electron microscopy (EM) to map true synaptic contacts. Numerous studies have described the ultrastructural characteristics and organization of hippocampal synapses in experimental animals (*Bourne and Harris, 2012*). However, there is very little information about the synaptic organization of the human hippocampus and the brain in general, which is a major problem since the question remains as to how much of the animal model information can be reliably extrapolated to humans. The majority of these studies are performed in specimens removed during the course of neurosurgery in patients with tumors or intractable epilepsy (*Alonso-Nanclares et al., 2008*; *Alonso-Nanclares et al., 2011*; *Androuin et al., 2018*; *Witcher et al., 2010*; *Yakoubi et al., 2019a*; *Yakoubi et al., 2019b*). Since it is inevitable that surgical excisions pass through cortical regions that are normal, this represents an excellent opportunity to study human brain material. The problem is that although this material is thought to be close to what would be expected in the normal brain, the results cannot be unequivocally considered as representative of the normal condition of the human brain. Thus, a major goal in neuroscience is to

**eLife digest** There are billions of nerve cells or neurons in the human brain, and each one can form thousands of connections, also called synapses, with other neurons. That means there are trillions of synapses in the brain that keep information flowing.

Studying the arrangement of individual neurons in the human brain, and the connections between them, is incredibly difficult because of its complexity. Scientists have tools that can image the whole brain and can measure the activity in different regions, but these tools only visualize brain structures that are large enough to be seen with human eyes. Synapses are much smaller (in the range of nanometers), and can only be seen using thin slices of preserved brain tissue through a technique called electron microscopy.

The hippocampus is a part of the human brain that is critical for memory, learning and spatial orientation, and is affected in epilepsy and Alzheimer's disease. Although numerous studies of the hippocampus have been performed in laboratory animals, such as mice, the question remains as to how much of the information gained from these studies applies to humans. Thus, studying the human brain directly is a major goal in neuroscience. However, the scarcity of human brain tissue suitable for the study of synapses is one of the most important issues to overcome. Fortunately, healthy human brain tissue that can be studied using electron microscopy is sometimes donated after death. Using these donations could improve the understanding of the synapses in normal brains and possible changes associated with disease.

Now, Montero-Crespo et al. have mapped synapses in the normal human hippocampus in three dimensions – providing the first detailed description of synaptic structure in this part of the brain. Using high-powered electron microscopes and donated brain tissue samples collected after death, Montero-Crespo et al. imaged almost 25,000 connections between neurons. The analysis showed that synapses were more densely packed in some layers of the hippocampus than in others. Most synapses were found to be connected to tiny dendritic 'spines' that sprout from dendritic branches of the neuron, and they activated (not suppressed) the next neuron.

Beyond its implications for better understanding of brain health and disease, this work could also advance computer modelling attempts to mimic the structure of the brain and its activity.

---

directly study human brain with no recorded neurological or psychiatric alterations. In the present study, we started to address the issue of the hippocampal synaptic organization by focusing on the CA1 field. This hippocampal field receives and integrates a massive amount of information in a laminar-specific manner, and sends projections mainly to the subiculum and to extrahippocampal subcortical nuclei and polymodal association cortices (*Amaral and Lavenex, 2007*).

Studying the human brain via EM techniques presents certain problems and the scarcity of human brain tissue that is suitable for the study of synaptic circuitry is one of the most important issues to overcome. Recently, we have shown that the 3D reconstruction method using Focused Ion Beam/Scanning Electron Microscopy (FIB/SEM) can be applied to study in detail the synaptic organization of the human brain obtained from autopsies, yielding excellent results (*Domínguez-Álvaro et al., 2018*; *Domínguez-Álvaro et al., 2019*).

For these reasons, we used FIB/SEM technology to perform a 3D analysis of the synaptic organization in the neuropil in all layers of the CA1 region from five human brain autopsies with a short postmortem delay. Specifically, we studied a variety of synaptic structural parameters including the synaptic density and spatial distribution, type of synapses, postsynaptic targets and the shape and size of the synaptic junctions.

The data reported in the present work constitutes the first extensive description of the synaptic organization in the human hippocampal CA1 field, which is a necessary step for better understanding its functional organization in health and disease.

## Results

We used coronal sections of the human hippocampus at the level of the hippocampal body and examined the CA1 field at both light and EM levels. Following a deep to superficial axis, the following main CA1 layers were analyzed: the alveus, *stratum oriens* (SO), *stratum pyramidale* (SP), *stratum*

*radiatum* (SR) and *stratum lacunosum-moleculare* (SLM) (*Figure 1—figure supplement 1*). Additionally, SP was subdivided into a deep part (dSP) close to the SO, and a superficial part (sSP), close to the SR.

## Light microscopy: volume fraction occupied by cortical elements

First, we estimated the total thickness of the CA1 field —including the alveus— in the radial axis. The average thickness was 2.70 ± 0.62 mm. Following a deep-superficial axis, the average length of each layer was: 0.34 ± 0.12 in the alveus; 0.06 ± 0.03 mm in SO; 1.13 ± 0.33 mm in SP; 0.55 ± 0.31 mm in SR; and 0.62 ± 0.16 mm in SLM. Thus, in relative terms, SP contributed the most to the total CA1 thickness (42%) followed by SLM (23%) then SR (20%), the alveus (13%) and SO (2%) (*Figure 1b*, *Supplementary file 1A*). We then assessed the cellular composition of every CA1 layer, including the volume fraction ($V_v$) occupied by different cortical elements (i.e., blood vessels, glial and neuronal somata and neuropil), estimated by applying the Cavalieri principle (*Gundersen et al., 1988*). The neuropil constituted undoubtedly the main element in all layers (more than 90%; *Figure 1—figure supplement 2b,f*, *Supplementary file 1A*) followed by blood vessels (range from 4.79% in SR to 7.58% in SO; *Figure 1—figure supplement 2b,c*, *Supplementary file 1A*). The volume fraction occupied by glial cell and neuronal bodies was less than 2% (*Figure 1—figure supplement 2b,d,e*, *Supplementary file 1A*), except for SP, where neuronal cell bodies occupied a volume of 4.23 ± 1.07% (*Figure 1—figure supplement 2b,e*, *Supplementary file 1A*). As expected, the volume occupied by neurons was significantly higher in SP than in any other layer (ANOVA, p<0.001). The neuropil was significantly more abundant in SR (94.19 ± 1.17%) than in SP (90.11 ± 1.32%, ANOVA, p=0.015) and SO (90.01 ± 3.07%; ANOVA, p=0.012). No further significant differences regarding cortical elements were found between any other layers.

## Electron microscopy

Each single reconstructed synapse was sorted according to different qualitative and quantitative parameters (see Material and Methods). Specifically, regarding qualitative characteristics, we distinguished four different parameters: i) the type of synapses: asymmetric synapses (AS) or symmetric synapses (SS); ii) the postsynaptic targets: axospinous (on the head or neck of the dendritic spine) or axodendritic (on spiny or aspiny dendritic shafts); and iii) the synaptic shape: macular, horseshoe-shaped, perforated or fragmented synapses. Additionally, three quantitative parameters were used for classification: i) the synaptic apposition surface (SAS) area, ii) SAS perimeter and iii) SAS curvature.

## Distribution of synapses in the neuropil

### Synaptic density

All synapses (n = 24,752) in the 75 stacks of images examined were fully reconstructed. After discarding the synapses not included in the unbiased counting frame (CF), a total of 19,269 synapses (AS = 18,138; SS = 1,131) were further considered for analysis and classification. The number of synapses per volume unit in every layer was calculated (synaptic density). The mean synaptic density was 0.67 ± 0.21 synapses/$\mu m^3$ (*Table 1*). Differences in synaptic density between layers were observed (*Figure 2a*; *Table 1*); sSP was the layer with the highest number of synapses per volume unit (0.99 ± 0.18 synapses/$\mu m^3$), whereas SO had the lowest synaptic density (0.45 ± 0.19 synapses/$\mu m^3$). However, synaptic density differences were only statistically significant between sSP and both SO (ANOVA, p=0.0005) and SLM (0.52 ± 0.08 synapses/$\mu m^3$; ANOVA, p=0.002; *Figure 2a*).

### Spatial distribution

Synapses fitted into a random spatial distribution in all layers since the observed F, G and K functions laid within the envelope generated by 99 simulations of the CSR model (*Anton-Sanchez et al., 2014*; *Merchán-Pérez et al., 2014*; *Figure 2—figure supplement 1*).

Furthermore, significant differences in the average intersynaptic distance were only found between sSP (604.00 ± 38.08 nm) and SO (742.81 ± 63.06 nm, ANOVA, p=0.0027; *Figure 2b*; *Table 1*). The maximum value was found in SO, whereas the minimum value was observed in sSP (*Figure 2b*; *Table 1*). Moreover, the variables synaptic density and intersynaptic distance were strongly and indirectly correlated ($R^2$ = 0.90).

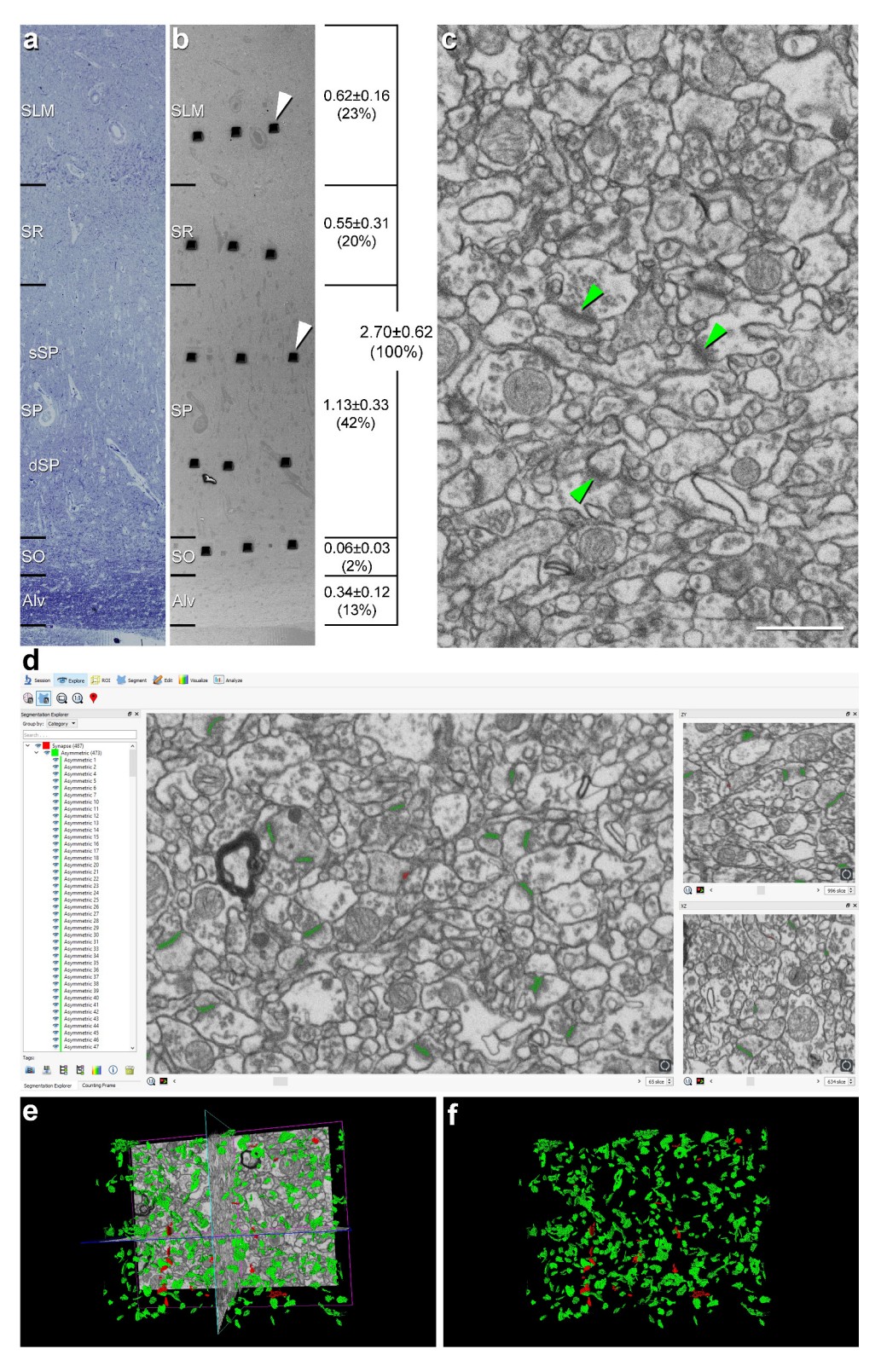

**Figure 1.** Correlative light/electron microscopy analysis of CA1 using FIB/SEM and EspINA software. (a, b) Delimitation of layers is based on the staining pattern of 1 μm thick semithin section stained with toluidine blue (a). This section is adjacent to the block surface (b), which is visualized with the SEM. This allows the exact location of the region of interest to be determined. The thickness of each stratum (mm; mean ± SD), as well as its relative contribution to the total CA1 thickness, is shown on the right side of panel (b). White arrowheads in (b) point to two of the trenches made in the

*Figure 1 continued on next page*

*Figure 1 continued*

neuropil (three per layer). (**c**), FIB/SEM image at a magnification of 5 nm/pixel. Some asymmetric synapses (AS) have been marked with green arrowheads. (**d**) Screenshot from the EspINA software interface. The stacks of images are visualized with EspINA software, permitting the identification and 3D reconstruction of all synapses in all spatial plans (XY, XZ and YZ). (**e**) Shows the three orthogonal planes and the 3D reconstruction of segmented synapses. (**f**) Only the segmented synapses are shown. AS are colored in green and symmetric synapses (SS) in red.

Alv: alveus; SO: *stratum oriens*; SP: *stratum pyramidale*; SR: *stratum radiatum*; SLM: *stratum lacunosum-moleculare*. See related ***Figure 1—figure supplements 1*** and ***2*** for further information. Scale bar in (**c**) corresponds to: 170 µm in **a**−**b**; 1 µm in (**c**).

The online version of this article includes the following figure supplement(s) for figure 1:

**Figure supplement 1.** Coronal section of the human hippocampus at the level of the hippocampal body.

**Figure supplement 2.** Stereological estimation of the volume occupied by different cortical elements in the CA1 using a stereological grid.

## Proportion of AS and SS

It is well established that AS are mostly glutamatergic and excitatory, whereas SS mostly GABAergic and inhibitory (***Ascoli et al., 2008***). Therefore, the proportions of AS and SS were calculated in each layer. Since synaptic junctions were fully reconstructed in the present study, all of them could be classified as AS or SS based on the thickness of their PSDs (***Merchán-Pérez et al., 2009***; ***Figure 3***).

The AS:SS ratio was close to 95:5 in all layers, except SLM, where the percentages were close to 90:10 (***Figure 2c***; ***Table 1***). We found significant differences in the proportion of excitatory and

**Table 1.** Data regarding synapses in all layers of the CA1.

Data in parentheses are not corrected with the shrinkage factor. AS: asymmetric synapse; CF: counting frame; SAS: synaptic apposition surface; SD: standard deviation; SO: *stratum oriens*; dSP: deep *stratum pyramidale*; sSP: superficial *stratum pyramidale*; SR: *stratum radiatum*; SLM: *stratum lacunosum-moleculare*; SS: symmetric synapse.

| | SO | dSP | sSP | SR | SLM | All layers |
|---|---|---|---|---|---|---|
| No. AS | 2,648 | 3,849 | 5,183 | 3,836 | 2,622 | 18,138 |
| No. SS | 166 | 281 | 196 | 172 | 316 | 1,131 |
| No. synapses (AS+SS) | 2,814 | 4,130 | 5,379 | 4,008 | 2,938 | 19,269 |
| % AS | 94.10% | 93.20% | 96.36% | 95.71% | 89.24% | 94.13% |
| % SS | 5.90% | 6.80% | 3.64% | 4.29% | 10.76% | 5.87% |
| CF volume ($\mu m^3$) | 6,221 (5,878) | 6,004 (5,486) | 5,400 (5,260) | 6,007 (5,697) | 5,690 (5,295) | 29,322 (27,616) |
| No. AS/$\mu m^3$ (mean ± SD) | 0.43± 0.19 (0.45± 0.22) | 0.64± 0.22 (0.70± 0.29) | 0.96± 0.18 (0.98± 0.20) | 0.64± 0.19 (0.67± 0.22) | 0.46± 0.07 (0.49± 0.10) | 0.63± 0.21 (0.66± 0.21) |
| No. SS/$\mu m^3$ (mean ± SD) | 0.03± 0.01 (0.03± 0.01) | 0.05± 0.02 (0.05± 0.02) | 0.04± 0.01 (0.04± 0.01) | 0.03± 0.01 (0.03± 0.01) | 0.06± 0.02 (0.06± 0.01) | 0.04± 0.01 (0.04± 0.01) |
| No. all synapses/$\mu m^3$ (mean ± SD) | 0.45± 0.19 (0.48± 0.23) | 0.69± 0.22 (0.75± 0.31) | 0.99± 0.18 (1.02± 0.19) | 0.67± 0.19 (0.70± 0.22) | 0.52± 0.08 (0.55± 0.11) | 0.67± 0.21 (0.70± 0.21) |
| Intersynaptic distance (nm; mean ± SD) | 742.81± 63.06 (717.55± 60.92) | 669.81± 54.18 (647.04± 52.34) | 604.00± 38.08 (583.46± 36.79) | 653.77± 69.51 (637.54± 67.15) | 689.65± 23.72 (666.20 22.91) | - |
| Area of SAS AS ($nm^2$; mean ± sem) | 86,716.52± 1,371.02 (80,906.52± 1,279.16) | 92,045.29± 1,192.92 (85,878.26± 1,112.99) | 88,061.63± 1,038.49 (82,161.50± 968.91) | 82,841.26± 1,201.47 (77,290.90± 1,120.97) | 91,419.95± 1,376.38 (85,294.81± 1,284.16) | 89,727.65± 5,775.90 (83,715.90± 5,388.91) |
| Area of SAS SS ($nm^2$; mean ± sem) | 85,737.60± 5,869.60 (79,993.18± 5,476.62) | 74,764.69± 3,057.33 (69,755.46± 2,852.49) | 58,305.43± 2,612.01 (54,398.67± 2,437.01) | 63,183.20± 2,734.96 (58,949.93± 2,551.72) | 57,390.19± 2,071.04 (53,545.05± 1,932.38) | 67,236.17± 4,456.52 (62,731.35± 4,157.93) |

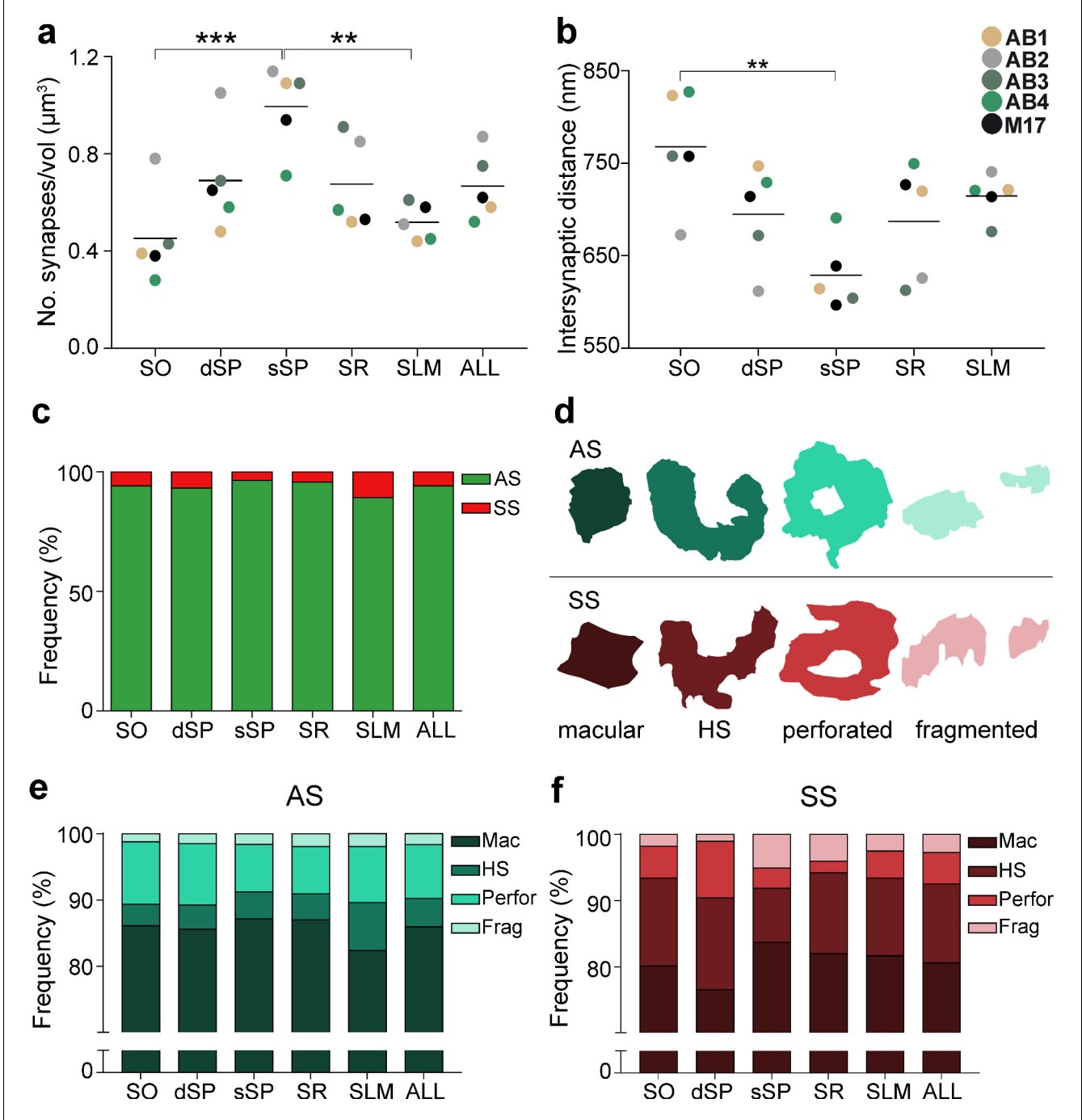

**Figure 2.** Synaptic density, intersynaptic distance, proportion of asymmetric synapses (AS) and symmetric synapses (SS), and proportion of synaptic shapes in CA1. (a) Graph showing the mean synaptic density in all layers. (b) Graph showing the mean intersynaptic distance in all layers. Each dot in (a) and (b) represents the data from each case, with the grey line showing the mean value. (c) Shows the percentages of AS and SS in all layers. (d) Illustrates examples of the different types of synapses based on the shape of the synaptic junction: macular, horseshoe-shaped (HS), perforated and fragmented. The upper and lower rows show examples of shapes of AS and SS, respectively. (e, f) Percentages of the different types of synaptic shapes within the population of AS (e) and SS (f) in all layers. SO: *stratum oriens*; dSP: deep *stratum pyramidale*; sSP: superficial *stratum pyramidale*; SR: *stratum radiatum*; SLM: *stratum lacunosum-moleculare*. **p<0.01; ***p<0.001. See related *Figure 2—figure supplements 1* and *2* for further information.

The online version of this article includes the following figure supplement(s) for figure 2:

**Figure supplement 1.** Analysis of the synaptic spatial distribution in the neuropil.

**Figure supplement 2.** Frequency plots for every type of synaptic shape found among axospinous AS and axodendritic AS.

inhibitory contacts between layers ($\chi^2$, p<0.0001). Specifically, the frequency of AS was significantly lower in dSP (93.20%) as compared to sSP (96.36%; $\chi^2$, p=4.381$\times10^{-12}$) and SR (95.71%; $\chi^2$, p=7.478$\times10^{-7}$; *Figure 2c*; *Table 1*). Furthermore, the proportion of SS was significantly higher in SLM than in any other layer (10.76%; $\chi^2$, p<0.0001, *Table 1*).

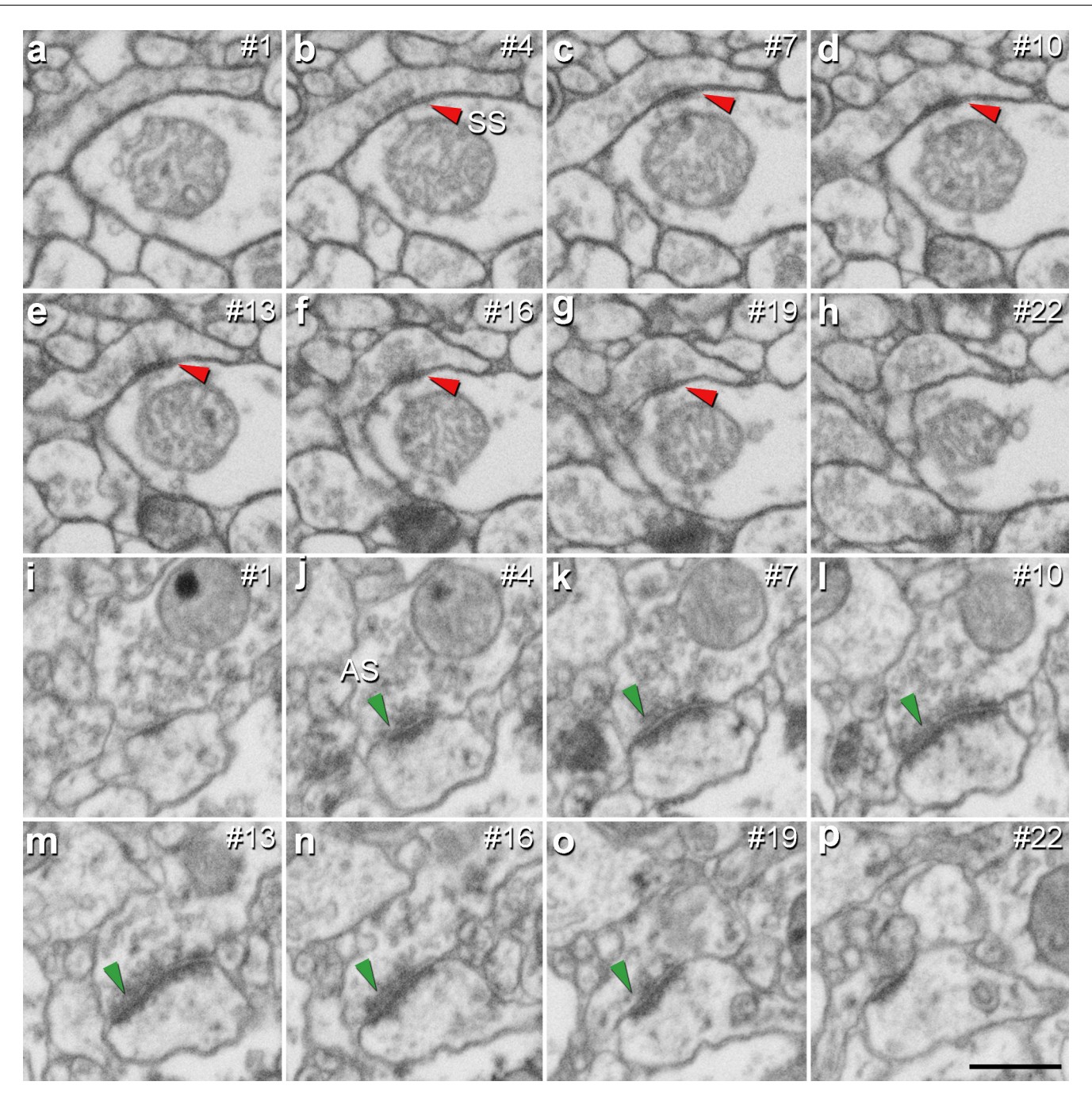

**Figure 3.** Identification of an asymmetric synapse (AS) and a symmetric synapse (SS) in the neuropil of the human CA1 region. (a−h) Crops from electron microscopy serial sections obtained by FIB/SEM to illustrate an SS (red arrowhead). (i−p) Crops from electron microscopy images following an AS (green arrowhead). The number of the section is indicated in the top right hand corner of each section, with a 60 nm thickness separation between images. Synapse classification was based on the examination of the full sequence of serial material. Scale bar in (p) corresponds to: 500 nm in a−p.

## Postsynaptic targets

Two main postsynaptic targets were considered (*Figure 4*): dendritic spines (axospinous synapses) and dendritic shafts (axodendritic synapses). In the case of axospinous synapses, the exact location of the synaptic contact was determined (i.e., the head or neck of the dendritic spine, *Figure 4a-k*). For axodendritic synapses, dendritic shafts were further classified as spiny (when dendritic spines could be observed emerging from the shaft) or aspiny. Only synapses whose postsynaptic target was clearly identifiable after navigation through the stack of images (n = 9,442; AS = 8,449, SS = 993) were considered for analysis.

## Total synaptic population

Despite the great disparity between layers, most synapses (AS+SS) were established on dendritic spines —especially on the head— (n = 7,469; 79.10%, ranging from 59.12% in SLM to 88.29% in sSP, *Supplementary file 1B*), rather than on dendritic shafts (n = 1,973; 20.90%, ranging from 11.71% in sSP to 40.88% in SLM, *Supplementary file 1B*). Synapses (AS+SS) on spiny shafts were more abundant than synapses on aspiny shafts in all CA1 layers, except for SLM ($\chi^2$, p<0.0001, *Supplementary file 1B*).

As a whole, axospinous AS were clearly the most abundant type of synapses in all layers (n = 7,369; 78.04%, ranging from 56.80% in SLM to 87.61% in sSP, *Figure 5*, *Figure 5—figure supplement 1*; *Supplementary file 1B*), followed by axodendritic AS, except for sSP (n = 1,080; 11.44%, ranging from 5.25% in sSP to 22.42% in SLM; *Figure 5*, *Figure 5—figure supplement 1*; *Supplementary file 1B*), where axodendritic SS were the second most abundant type of synapses (n = 893; 9.46%, ranging from 6.46% in sSP to 18.46% in SLM; *Figure 5*, *Figure 5—figure supplement 1*; *Supplementary file 1B*). Axospinous SS were remarkably scarce (n = 100; 1.06%, ranging from 0.37% in SR to 2.32% in SLM; *Figure 5*, *Figure 5—figure supplement 1*; *Supplementary file 1B*).

Significant differences in the proportion of synapses were found between layers ($\chi^2$, p<0.0001) (*Supplementary file 1C*). Both axodendritic AS and axodendritic SS were clearly more frequent in SLM than in any other layer ($\chi^2$, p<0.0001). sSP presented the largest proportion of axospinous AS ($\chi^2$, p<0.0001) and the lowest frequency of axodendritic AS ($\chi^2$, p<0.0001). Additionally, a lower prevalence of axospinous AS and a larger proportion of axodendritic AS were observed in SO compared to dSP ($\chi^2$, p=0.0004 and p=1.518×10$^{-7}$, respectively) (*Supplementary files 1D-G*). Finally, the prevalence of axospinous SS was significantly higher in dSP and SLM than in any other layer ($\chi^2$, p<0.001 in dSP vs SO and dSP vs sSP; p<0.0001 in the rest of the cases).

## Postsynaptic preference of AS and SS

Regardless of the layer, most AS were established on dendritic spines (n = 7,369; 87.22%, ranging from 71.70% in SLM to 94.34% in sSP in the population of AS; *Figure 4l*; *Supplementary file 1H*), and they were found almost exclusively on the head of the spines (>99.5% in all layers, *Figure 5*). The remaining AS were established on dendritic shafts (n = 1,080; 12.78%, ranging from 5.66% in sSP to 28.30% in SLM; *Figure 4l*; *Supplementary file 1H*), with a preference for spiny shafts in SO and sSP, whereas in SLM the preference was for aspiny shafts (*Figure 5*). In the case of SS, most were axodendritic (n = 893; 89.93%), ranging from 82.63% in dSP to 96.36% in SR (*Figure 4m*; *Supplementary file 1H*). SS showed a clear preference for spiny shafts in all layers except in SLM (*Figure 5*). The remaining SS were established on dendritic spines (n = 100; 10.07%, ranging from 3.64% in SR to 17.37% in dSP; *Figure 4m*; *Supplementary file 1H*). These axospinous SS were found especially on the head of the spines (82%, *Figure 5*).

In every layer, we found a consistent association for AS and dendritic spines, and for SS and dendritic shafts ($\chi^2$, p<0.0001). Moreover, the preference of inhibitory contacts for dendritic shafts was found for both spiny and aspiny dendritic shafts, regardless of the layer ($\chi^2$, p<0.0001). Spiny shafts received a higher proportion of SS than AS in all layers, except for SO, while aspiny shafts received a higher proportion of AS than SS in all layers, especially in dSP.

Although scarce, dendritic spines receiving multiple synapses were found in all layers (2.11% of total spines in all layers; *Figure 5—figure supplement 1c*; *Supplementary file 1I*), whereas single axospinous SS were extremely rare or even not found in some layers (*Figure 5—figure supplement*

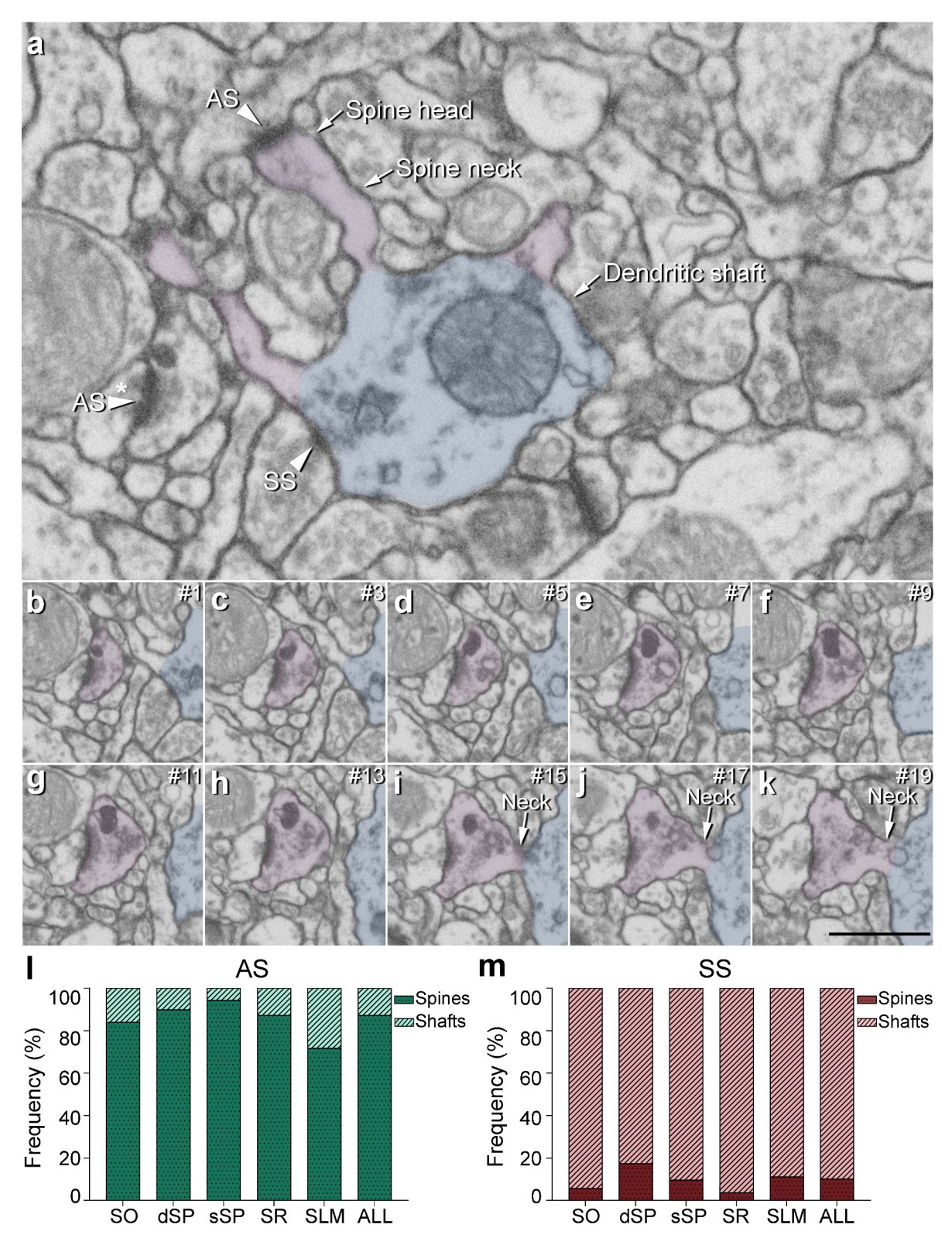

**Figure 4.** Postsynaptic target identification in serial electron microscopy images. (a) A crop from an electron microscopy section obtained by FIB/SEM to illustrate a dendritic shaft (blue) with three dendritic spines (purple) emerging from the shaft (the neck and head have been indicated in one of the spines). A symmetric synapse (SS) on the dendritic shaft is pointed out with an arrowhead. An axospinous asymmetric synapse (AS) (marked with an arrowhead) is established on the head of one of the spines. Another AS is indicated (arrowhead with asterisk); however, the nature of the postsynaptic

*Figure 4 continued on next page*

*Figure 4 continued*

element where the synapse is established cannot be distinguished in a single section. (**b**−**k**) Crops from electron microscopy serial sections to illustrate the nature of the postsynaptic element of the AS (arrowhead with asterisk) in (**a**). By following up from this AS through the stack of images (the number of the section is indicated in the top right hand corner of each section; 40 nm thickness separation between images), a dendritic spine (purple), whose neck has been labeled emerging from the dendritic shaft (blue), can be unequivocally identified. (**l**, **m**) The percentage of axospinous and axodendritic synapses within the AS (**l**) and SS (**m**) populations in all layers of CA1. SO: *stratum oriens*; dSP: deep *stratum pyramidale*; sSP: superficial *stratum pyramidale*; SR: *stratum radiatum*; SLM: *stratum lacunosum-moleculare*. Scale bar in (**k**) corresponds to: 1 μm in (**a**); 500 nm in (**b**−**k**).

---

*1c*; *Supplementary file 1I*). Moreover, multiple-headed dendritic spines (double-headed in most cases) were also observed (1.39% of total spines in all layers; *Supplementary file 1I*).

## Shape of the synaptic junctions

Synapses were categorized as macular, horseshoe-shaped, perforated or fragmented (n = 19,269; AS = 18,138, SS = 1,131; *Figure 2d-f*). The vast majority of both AS and SS (more than 75% in all layers) had a macular shape (85.95% and 80.55%, respectively; *Figure 2e,f*; *Supplementary file 1J*), followed by perforated synapses in the case of AS (8.13%, *Figure 2e,f*; *Supplementary file 1J*) and horseshoe-shaped synapses in the case of SS (11.94%, *Figure 2e,f*; *Supplementary file 1J*). We observed that some synaptic shapes were more prevalent in some layers. Overall, AS with complex shapes (that is, including either horseshoe-shaped, perforated or fragmented) were more abundant in SLM than in any other layer ($\chi^2$, p<0.001; *Figure 2e*), especially horseshoe-shaped synapses ($\chi^2$, p<0.0001; *Figure 2e*). They were mainly located in dendritic shafts ($\chi^2$, p=1.259×10$^{-5}$; *Figure 2—figure supplement 2e*; *Supplementary file 1K*). Additionally, perforated AS were observed more frequently in SO and dSP than in sSP and SR ($\chi^2$, p<0.001; *Figure 2e*). No differences could be observed in the case of SS ($\chi^2$, p>0.001).

Considering both AS and SS against the four types of synaptic shapes in each layer, we found that horseshoe-shaped synapses were significantly more abundant in the SS population than in the AS population in all layers ($\chi^2$, p<0.001). No synaptic shape was more frequent among AS.

## Size of the synapses

### SAS area and perimeter

Morphological features of SAS were extracted with EspINA software for both AS and SS (n = 19,269; AS = 18,138, SS = 1,131; *Figure 6*, *Figure 6—figure supplements 1*, *2*, *3*; *Supplementary files 1L-N*).

The mean SAS areas of AS and SS were 89,727.65 nm$^2$ and 67,236.17 nm$^2$, respectively, while the mean SAS perimeters of AS and SS were 1,458.82 and 1,378.38 nm, respectively (*Table 1*; *Supplementary file 1L*). No differences were observed between layers regarding the size (area and perimeter) of the synapses both for AS (ANOVA, p>0.05; *Figure 6a*; *Supplementary file 1L*) and SS (ANOVA, p>0.05; *Figure 6a*; *Supplementary file 1L*). However, AS had significantly larger areas than SS when considering all synapses together (MW, p=0.032; *Figure 6a*; *Supplementary file 1L*), but when focusing on particular layers, this difference in area between AS and SS was only observed in dSP, sSP and SLM (MW, p=0.016, p=0.032 and p=0.008, respectively; *Figure 6a*; *Supplementary file 1L*). No differences were found in perimeter measurements (MW, p>0.05). Although significant differences in SAS area did not extrapolate to differences in SAS perimeter, there was a strong correlation between these two parameters ($R^2$ = 0.81 for all synapses; $R^2$ = 0.82 for AS; $R^2$ = 0.81 for SS).

To further characterize the size distribution of the SAS of both AS and SS, we plotted the frequency histograms of SAS areas for each individual layer and all layers. Frequency histograms had similar shapes for both types of synapses when considering all layers and within each layer, with a positive skewness (that is, most synapses presented small SAS area values). Moreover, the frequency distributions of AS and SS greatly overlapped, as did the frequency distributions of SAS area between the layers (KS, p>0.001; *Figure 6b*, *Figure 6—figure supplement 1a, b*). Furthermore, we found that both types of synapses (AS and SS) can be fitted to log-normal or log-logistic probability density functions. These distributions, with some variations in the parameters of the functions

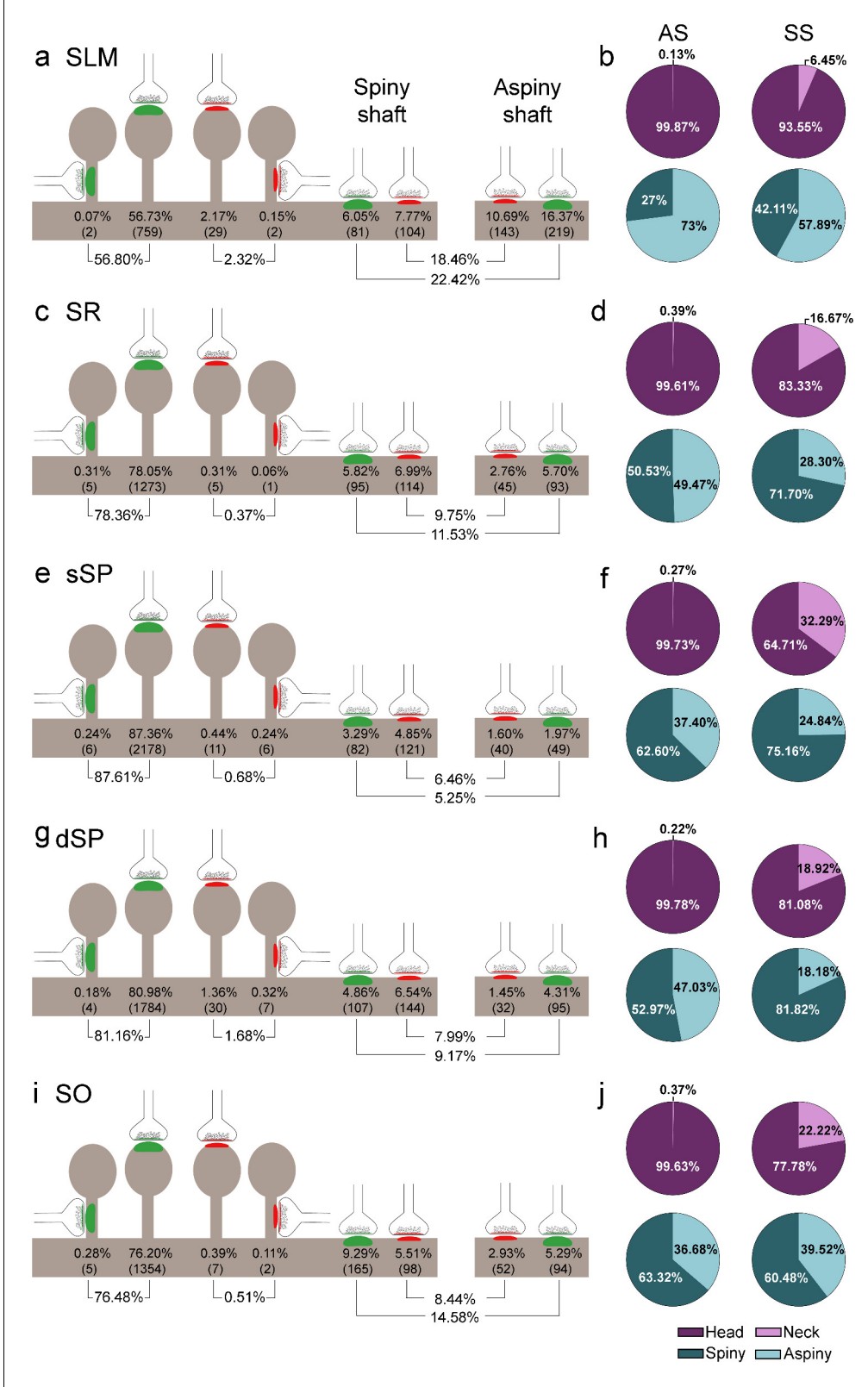

**Figure 5.** Representation of the distribution of synapses according to their postsynaptic targets in all layers of CA1. (a, c, e, g, i) Shows the percentages of axospinous (both on the head and the neck of dendritic spines) and axodendritic (both on spiny and aspiny shafts) asymmetric synapses (AS; green) and symmetric synapses (SS; red). The numbers of each synaptic type are shown in brackets. (b, d, f, h, j) Pie charts to illustrate the proportions of AS and SS according to their location as axospinous synapses (i.e., on the head or on the neck of the spine) or axodendritic synapses (i.

*Figure 5 continued on next page*

*Figure 5 continued*

e., spiny or aspiny shafts). SO: *stratum oriens*; dSP: deep *stratum pyramidale*; sSP: superficial *stratum pyramidale*; SR: *stratum radiatum*; SLM: *stratum lacunosum-moleculare*. See related *Figure 5—figure supplement 1* for further information.

The online version of this article includes the following figure supplement(s) for figure 5:

**Figure supplement 1.** Schematic representation of the distribution of synapses regarding postsynaptic targets and dendritic spines in the whole CA1.

(*Supplementary file 1L*), were found in each layer and the whole CA1 (all layers pooled together) for both AS and SS (*Supplementary file 1L*; *Figure 6—figure supplement 2*).

Additionally, we studied synaptic size regarding the postsynaptic targets. Although both the mean SAS area and the perimeter of axodendritic AS were larger (117,360.02 nm$^2$ and 1,686.99 nm, respectively) than axospinous AS (98,200.61 nm$^2$ and 1,548.38 nm, respectively), these differences were not statistically significant (MW, p>0.05; *Figure 6c*; *Supplementary file 1M*). Only axodendritic AS in SLM were significantly larger than axospinous AS regarding both SAS area and perimeter (MW, p=0.008 for area, p=0.016 for perimeter; *Figure 6c*; *Supplementary file 1M*). Overall, axodendritic SS (71,218.23 nm$^2$) had a larger mean area than axospinous SS (49,044.59 nm$^2$) (*Figure 6d*; *Supplementary file 1M*) but, again, this difference in the mean SAS area was significant only in SLM (MW, p=0.032; *Figure 6d*; *Supplementary file 1M*).

Analyses were carried out to determine the differences in synaptic size in terms of the shape of the synaptic junctions. Macular synapses were smaller than the rest of the more complex-shaped synapses for both AS (mean macular SAS area: 70,322.92 nm$^2$, mean complex-shaped SAS area: 200,539.32 nm$^2$) and SS (mean macular SAS area: 56,769.81 nm$^2$, mean complex-shaped SAS area: 115,170.07 nm$^2$). However, these differences were only significant in the case of AS, as demonstrated by both mean SAS area and perimeter (ANOVA, p<0.0001 in all cases, except for macular AS and horseshoe-shaped AS, p=0.002; *Figure 6e,f*; *Supplementary file 1N*). This difference was also observed between AS in all layers (ANOVA, p<0.05; *Figure 6e*; *Supplementary file 1N*). No differences were observed in the synaptic size of the different synaptic shapes between the layers (ANOVA, p>0.05).

## SAS curvature

While synaptic size parameters area and perimeter were highly correlated ($R^2$ = 0.81 for all synapses; $R^2$ = 0.82 for AS; $R^2$ = 0.81 for SS), curvature measurements showed very little association with either area ($R^2$ = 0.05 for all synapses; $R^2$ = 0.05 for AS; $R^2$ = 0.00 for SS) or perimeter ($R^2$ = 0.08 for all synapses; $R^2$ = 0.09 for AS; $R^2$ = 0.00 for SS). Consequently, differences observed in these two parameters did not extrapolate to variations in the curvature (*Figure 6—figure supplement 3*).

No differences in the curvature of the synapses were observed between AS and SS (mean SAS curvature of AS: 0.050; mean SAS curvature of SS: 0.047; *Figure 6—figure supplement 3a*; *Supplementary file 1L*). Likewise, no curvature differences were seen between the layers (ANOVA, p>0.05; *Figure 6—figure supplement 3a*).

The frequency histograms of SAS curvature ratios showed a positive skewness with a greater proportion of synapses presenting lower values, meaning a larger prevalence of flatter synapses than more curved ones for both AS and SS populations, but also within every layer, with great overlap among all the distributions (KS, p>0.001; *Figure 6—figure supplements 1c,d* and *3b*).

Regarding the postsynaptic targets, the curvature ratio of axospinous and axodendritic synapses did not differ for either population — AS or SS (MW, p>0.05; *Figure 6—figure supplement 3c,d*; *Supplementary file 1M*).

When focusing on the shape of the synaptic junction, fragmented AS were found to be more curved than macular AS (ANOVA, p=0.04) — a difference that was maintained through all layers (ANOVA, p<0.001; *Figure 6—figure supplement 3e*; *Supplementary file 1N*). In the case of SS, fragmented SS were observed to be more curved than the rest of the synaptic shape types in SLM (ANOVA, p=0.0480 for macular SS-fragmented SS; p=0.0050 for HS SS-fragmented SS; and p=0.0009 for perforated SS-fragmented SS; *Figure 6—figure supplement 3f*; *Supplementary file 1N*). Differences in the mean SAS curvature were also observed within the same synaptic shape type between layers. In this regard, SLM presented flatter horseshoe-shaped AS than both dSP (ANOVA, p=0.0035) and sSP (p=0.0005), while SO exhibited flatter perforated AS than sSP (ANOVA,

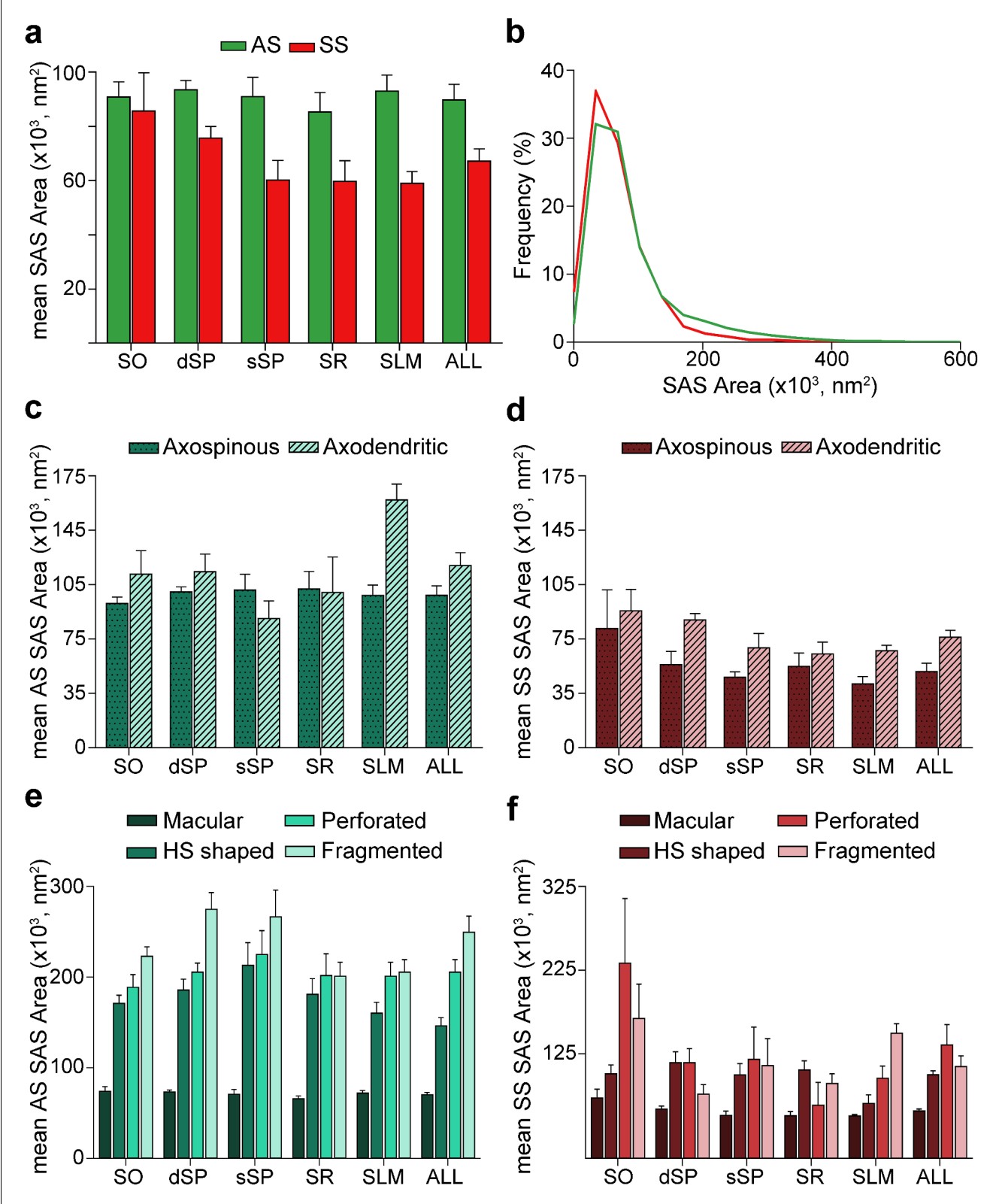

**Figure 6.** Synaptic apposition surface (SAS) area measurements from 5 subjects. (a) Mean SAS area of asymmetric synapses (AS; green) and symmetric synapses (SS; red) are represented for each layer of CA1 (mean ± sem). (b) Frequency distribution of SAS areas for both AS (green line; n = 18,138 synapses) and SS (red line; n = 1,131 synapses) in all layers of CA1. No differences were observed in the frequency distribution of SAS areas between the two synaptic types (KS, p>0.05). (c), (d) Mean SAS area of axospinous and axodendritic synapses are also shown for AS (c) and SS (d) in the whole
*Figure 6 continued on next page*

*Figure 6 continued*

CA1 and per layer (mean ± sem). Both axodendritic AS and SS were larger in SLM than axospinous AS (MW, p<0.01) and SS (MW, p<0.05), respectively, while in the rest of the layers, no differences were observed. (e), (f) Mean SAS area related to the different synaptic shapes are plotted for both AS (e) and SS (f) in all layers of CA1 (mean ± sem). Macular synapses are significantly smaller than the other more complex-shaped ones (i.e., horseshoe-shaped (HS), perforated and fragmented); however, this difference is only significant for AS (ANOVA, p<0.0001). SO: *stratum oriens*; dSP: deep *stratum pyramidale*; sSP: superficial *stratum pyramidale*; SR: *stratum radiatum*; SLM: *stratum lacunosum-moleculare*. See related *Figure 6—figure supplements 1* and *2* for further information.

The online version of this article includes the following figure supplement(s) for figure 6:

**Figure supplement 1.** Frequency distribution histograms of the synaptic apposition surface (SAS) area and curvature in CA1.
**Figure supplement 2.** Frequency histograms of synaptic apposition surface (SAS) areas and their corresponding best-fit probability density functions.
**Figure supplement 3.** Synaptic size: synaptic apposition surface (SAS) curvature ratio from 5 subjects.

p=$1.246 \times 10^{-5}$) and SR (ANOVA, p=$5.538 \times 10^{-5}$; *Figure 6—figure supplement 3e*; *Supplementary file 1N*).

## Interindividual variability

Differences between cases were observed regarding several of the parameters examined in several layers (*Supplementary files 1O-AD*). All significant differences are reported under the corresponding tables for each individual case. Importantly, differences between individual cases were not necessarily found with respect to the same parameter, or in the same layer or in the same direction (increase or decrease). For instance, case AB1 presented a larger volume fraction of blood vessels in SO than the rest of the cases (ANOVA, p<0.05; *Supplementary file 1O*), except for case AB2. In case AB2, the volume fraction occupied by neuronal bodies in SR was higher than in the rest of the subjects, except for AB3 (ANOVA, p<0.01; *Supplementary file 1O*). Additionally, the volume occupied by glia in the SLM of case AB1 was higher than in the rest of the cases, except for AB3 (ANOVA, p<0.05; *Supplementary file 1O*). Furthermore, compared to the rest of the cases, AB2 and AB3 presented higher synaptic densities in SR (ANOVA, p<0.05; *Supplementary file 1S*). Also, compared to the rest of the subjects, AB2 exhibited a higher synaptic density in SO (ANOVA, p<0.01; *Supplementary file 1P*). In addition, the proportion of SS was higher in SLM in M17 than in AB1, AB2 and AB3 ($\chi^2$, p<0.001; *Supplementary file 1T*).

When focusing on postsynaptic targets, SLM was one of the layers with the greatest differences among cases ($\chi^2$, p=$1.000 \times 10^{-17}$; *Supplementary file 1Y*). In this layer, out of all the cases, case AB1 exhibited the highest proportion of axospinous AS and the lowest percentage of both axodendritic AS and SS ($\chi^2$, p<0.0001; *Supplementary file 1Y*). Additionally, a larger proportion of axospinous AS was also observed in subject AB3 when compared to cases AB2 and M17 ($\chi^2$, p<0.0001; *Supplementary file 1Y*).

Macular synaptic junctions were clearly the most abundant type in all cases and layers. However, perforated AS were especially abundant in subject AB4 compared to the rest of the cases in all layers ($\chi^2$, p<0.001; *Supplementary files 1Z-AD*) with the exception of AB1 in sSP. Additionally, in case AB4, the AS in sSP were larger than in the rest of the individuals (ANOVA, p<0.0001; *Supplementary file 1R*), apart from in the case of M17.

## Discussion

The present study constitutes the first exhaustive description of the synaptic organization in the neuropil of the human CA1 field using 3D EM. The following major results were obtained: (i) there are significant differences in the synaptic density between layers; (ii) synapses fitted into a random spatial distribution; (iii) most synapses are excitatory, targeting dendritic spines and displaying a macular shape, regardless of the layer — although significant differences were observed between certain layers; (iv) SLM showed several peculiarities compared with other layers, such as a larger proportion of inhibitory synapses, a higher prevalence of both AS and SS axodendritic synapses, and the presence of more complex synaptic shapes. The wide range of differences in the synaptic organization of the human CA1 layers found in the present study may be related to the variety of inputs arriving in a layer-dependent manner (*Figure 7*).

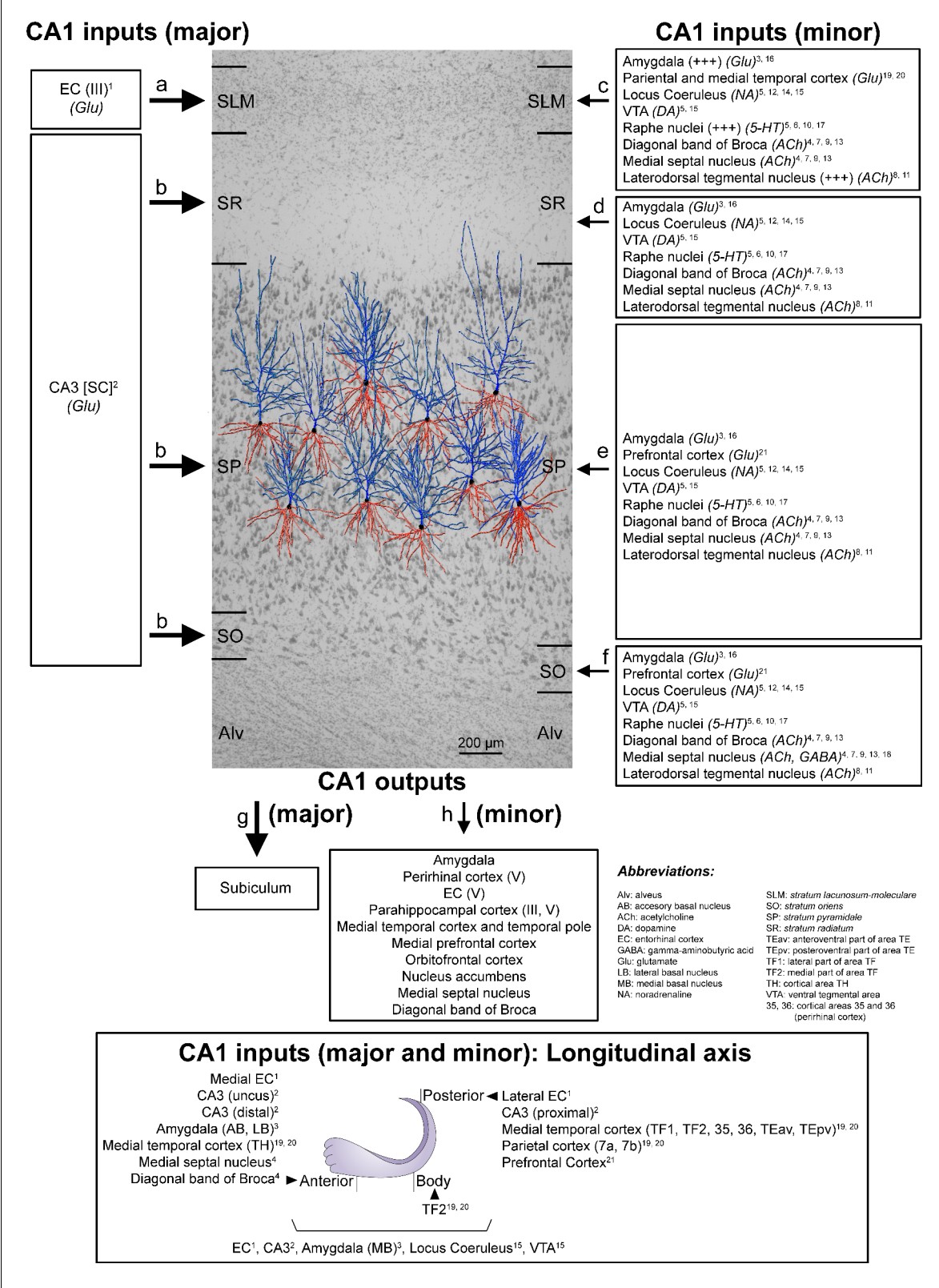

**Figure 7.** Schematic representation of the main direct connections between CA1 and other brain regions in primates (monkeys, unless otherwise specified; note that all abbreviations are defined in the figure itself). Photomicrograph of a Nissl-stained coronal brain section from the human CA1 (in black and white) with reconstructed pyramidal neurons (taken from *Benavides-Piccione et al., 2020*) superimposed on the same scale. The pyramidal neurons have been placed in the middle of the SP, approximately where they were injected with Lucifer Yellow. The apical and basal dendritic arbors

*Figure 7 continued on next page*

*Figure 7 continued*

are colored in blue and red, respectively. Major and minor projections have been represented with large and small arrows, respectively. See Appendix 1 for further information on human CA1 connectivity.

## CA1 structural composition

The neuropil represents the main structural component of CA1 (more than 90% of all layers). The contribution of SP to the total CA1 radial extension accounted for almost a half of the total thickness (SP thickness: 1.13 mm; total CA1 thickness: 2.70 mm). This great extent of SP represents a major difference with the rodent brain and other species. Indeed, important variances can be observed in the hippocampal neuroanatomy of humans compared to rodents (*Amaral and Lavenex, 2007*; *Benavides-Piccione et al., 2020*; *Duvernoy, 2005*; *Slomianka et al., 2011*; *Tapia-González et al., 2020*). In the rat hippocampus, SP is around five cell bodies thick and neuronal somata are densely packed, being SR the layer that contributes the most to the total CA1 thickness. In humans, SP can be up to 30 cell somata thick, with a wider separation of neurons compared to other species. As previously discussed in *Benavides-Piccione et al., 2020*, this sometimes refers to a 'corticalization' of the human CA1 pyramidal cell layer because it resembles a neocortical cytoarchitecture, which most probably has fundamental functional and hodological consequences: the basal and apical dendrites of human pyramidal cells are intermixed in the pyramidal cell layer (*Figure 7*), whereas in rodents, the basal and apical dendritic arbors are basically separated (basal dendrites in SO; apical dendrites in SR).

## Synaptic density

Synapses were found in all layers except the alveus where they were virtually nonexistent. The mean synaptic density was 0.67 synapses/$\mu m^3$. However, synaptic density was not homogenous among layers. Consistently in all individuals, the highest value was found in sSP (0.99 synapses/$\mu m^3$), followed by dSP (0.69 synapses/$\mu m^3$), while the lowest was observed in SO (0.45 synapses/$\mu m^3$). Since no quantitative 3D analysis of the synaptic organization in the human hippocampus has been performed before, our data could not be compared to previous reports. However, in recent studies from our group using FIB/SEM to analyze the synaptic density in the rodent CA1 field, the following values were obtained: 2.53 synapses/$\mu m^3$ in SO, 2.36 synapses/$\mu m^3$ in SR and 1.72 synapses/$\mu m^3$ in SLM in the mouse (*Santuy et al., 2020*), and 2.52 synapses/$\mu m^3$ in SR in the rat (*Blazquez-Llorca et al., 2020*). These values are much higher than the ones found in the present work for the human CA1 (*Table 1*). Such huge differences in synaptic density between humans and rodents — together with the above-mentioned divergences in the morphology and distribution of pyramidal cells in the SP of CA1 (*Benavides-Piccione et al., 2020*), as well as differences in other anatomical, genetic, molecular and physiological features (*Benavides-Piccione et al., 2020*; *Blazquez-Llorca et al., 2020*; *Ding, 2013*; *Hawrylycz et al., 2012*; *Santuy et al., 2020*; *Tapia-González et al., 2020*; *van Dijk et al., 2016*)— further support the notion that there are remarkable differences between the human and rodent CA1. These differences clearly need to be taken into consideration when making interpretations in translational studies comparing one species to another.

## Spatial synaptic distribution, proportion of synapses and postsynaptic targets

While synaptic density differed across layers, the spatial organization of synapses was consistently random in all layers. Randomly distributed synapses have also been described in the somatosensory cortex of rats and the frontal and transentorhinal cortices of the human brain (*Blazquez-Llorca et al., 2013*; *Domínguez-Álvaro et al., 2018*; *Merchán-Pérez et al., 2014*; *Santuy et al., 2018a*), suggesting that this synaptic characteristic is a widespread 'rule' of the cerebral cortex of different species.

It has also been consistently reported that the neuropil is characterized by a much higher number of excitatory contacts compared to inhibitory synapses in different brain regions and species (*Beaulieu and Colonnier, 1985*; *Bourne and Harris, 2012*; *DeFelipe, 2011*; *Domínguez-Álvaro et al., 2018*; *Megías et al., 2001*; *Santuy et al., 2018a*). In the present study, the density of

inhibitory synapses was particularly low in most CA1 layers (AS:SS ratio in all layers was around 95:5 except for in SLM, where the ratio was close to 90:10). This data is in line with our study using FIB/SEM to analyze the synaptic density in the mouse (where the proportion of synapses that were inhibitory was 8% in the SLM, and approximately 2% in the case of the SR and SO) (*Santuy et al., 2020*) and in the rat CA1 field (where 4% of the synapses in SR were inhibitory) (*Blazquez-Llorca et al., 2020*).

Regarding postsynaptic preferences, we observed a clear preference of excitatory axons and inhibitory axons for dendritic spines and dendritic shafts, respectively, which is also characteristic in other cortical regions and species, although variations in their percentages have been reported (*Beaulieu and Colonnier, 1985*; *Beaulieu et al., 1992*; *Bourne and Harris, 2012*; *Domínguez-Álvaro et al., 2019*; *Megías et al., 2001*; *Micheva and Beaulieu, 1996*; *Santuy et al., 2018a*). For example, axospinous AS are especially abundant in sSP (87.61%) when compared to other brain regions in both humans and other species such as layer II of the human transentorhinal cortex, where axospinous AS account for only 55% of the total synaptic population (*Domínguez-Álvaro et al., 2019*).

## Shape and size of the synapses

Most synapses presented a simple, macular shape (accounting for 86% of the synapses in all layers of CA1), in agreement with previous reports in different brain areas and species (*Domínguez-Álvaro et al., 2019*; *Geinisman et al., 1986*; *Jones and Calverley, 1991*; *Neuman et al., 2015*; *Santuy et al., 2018b*).

The shape and size of the synaptic junctions are strongly correlated with release probability, synaptic strength, efficacy and plasticity (*Biederer et al., 2017*; *Ganeshina et al., 2004a*; *Ganeshina et al., 2004b*; *Holderith et al., 2012*). In this regard, all three types of non-macular synapses (with more complex shapes) were larger than macular ones. Although the functional significance of perforations is still unclear, perforated synapses are known to have more AMPA and NMDA receptors than macular synapses and are thought to constitute a relatively powerful population of synapses with more long-lasting memory-related functionality than their smaller, macular counterparts (*Ganeshina et al., 2004a*; *Ganeshina et al., 2004b*; *Vincent-Lamarre et al., 2018*).

The size of both types of synaptic junctions (AS and SS) can be fitted to log-normal or log-logistic probability density functions (see *Figure 6—figure supplement 2*). These distributions show a characteristic skewed shape, with a long tail to the right (*Kumar and Kundu, 2009*). This is consistent with the fact that small macular synapses predominate, while larger horseshoe or perforated synapses are less common. Indeed, previous studies from our laboratory found that AS and SS followed a log-normal distribution in all layers of the rat somatosensory cortex (*Merchán-Pérez et al., 2014*; *Santuy et al., 2018b*). Interestingly, some other synaptic parameters —such as synaptic strength and spike transmission probability— follow log-normal distributions (for review, see *Buzsáki and Mizuseki, 2014*). For example, the distribution of the size of unitary excitatory postsynaptic potentials (EPSP) is very similar to the distribution of the size of SAS reported here (*Lefort et al., 2009*; *Song et al., 2005*). When analyzing the synaptic size distribution of pairs of synapses formed by the same axon, a quantized distribution of synaptic strengths yielding a similar skewed curve was observed in the rat hippocampus (*Bartol et al., 2015*; *Bromer et al., 2018*); since we did not trace the parental axon of each segmented synapse in our samples, it remains unclear whether a similar quantitation process occurs in the human hippocampus. Although the extraordinary diversity of excitatory synapse sizes is commonly attributed to activity-dependent processes that drive synaptic growth and diminution, recent studies also point to activity-independent size fluctuations, possibly driven by innate synaptic molecule dynamics, as important generators of size diversity. Specifically, activity-dependent processes seem to primarily dictate the scale rather than the shape of synaptic size distributions (*Hazan and Ziv, 2020*).

Considering all synapses, excitatory contacts were larger than inhibitory ones, as has also been observed in layer II of the human transentorhinal cortex (*Domínguez-Álvaro et al., 2018*); however, this contrasted with the findings in the somatosensory cortex (*Santuy et al., 2018b*) and SR of CA1 in the rat (*Blazquez-Llorca et al., 2020*). A tendency towards axodendritic synapses being bigger than axospinous synapses was also observed; however, this difference was only significant in the case of SLM synapses. Complex-shaped AS were also found more frequently associated with axodendritic AS than with axospinous AS in SLM, while the opposite was the case for the rest of the

layers. These findings agree with reports in the rat hippocampus, where excitatory synapses on SLM dendrites were observed to be: (i) larger than synapses in other layers; (ii) more frequently perforated (approximately 40%); and (iii) located to a greater extent on dendritic shafts (*Megías et al., 2001*).

### Relation between synaptic inputs and synaptic organization of each layer

The wide range of differences in the synaptic organization of the human CA1 layers found in the present study, especially between SLM and the rest of layers, may be related to the variety of inputs arriving to these layers (*Figure 7*). Unfortunately, detailed hippocampal human connectivity is far to be known: data directly obtained from human brains are very scarce and most data are inferred from rodents and primates (*Insausti and Amaral, 2012*; *Spruston and McBain, 2007*). In the primate brain, the CA1 field receives a wide variety of inputs from multiple subcortical and cortical brain regions (*Insausti and Amaral, 2012*; *Spruston and McBain, 2007*), being the major input to CA1 originated in the EC. Specifically, neurons located in layer III (and layer V) of the EC project directly to SLM, whereas neurons in layer II project to the rest of CA1 layers indirectly via the DG and CA3 field (*Insausti and Amaral, 2012*; *Kondo et al., 2009*). Considering both the synaptic data obtained in the present study and the connectivity knowledge in monkeys, it may seem that the synaptic organization in the layers receiving CA3 Schaffer collateral inputs (i.e. SO, SP and SR) differs with the synaptic organization found in the layer receiving direct inputs from the EC (i.e. SLM). Additionally, SLM receives a higher number of glutamatergic inputs from the amygdala and from the parietal and medial temporal cortex and higher numbers of serotonergic and Substance-P immunoreactive fibers, with a possible extrinsic origin in the Raphe nuclei and the laterodorsal tegmental nucleus (*Figure 7*).

It has been proposed that the CA3-CA1 synaptic connection plays a key role in the learning-induced synaptic potentiation of the hippocampus (*Whitlock et al., 2006*), while the direct projection from EC to SLM of CA1 seems to modulate information flow through the hippocampus (*Dvorak-Carbone and Schuman, 1999*). It has been reported that a high-frequency stimulation in SLM evokes an inhibition sufficiently strong to prevent CA1 pyramidal cells from spiking in response to Schaffer collaterals input (*Dvorak-Carbone and Schuman, 1999*). This finding could be supported by our present data showing an elevated inhibitory synapse ratio in comparison to other CA1 layers. It has also been described that afferents from the EC contact not only the apical tuft of CA1 pyramidal cells, but also interneurons of the SLM (*Lacaille and Schwartzkroin, 1988*). One of these interneurons are the neurogliaform cells, which receive monosynaptic inputs from the EC and are also synaptically coupled with each other and with CA1 pyramidal cells (*Capogna, 2011*). Whether the higher proportion of axodendritic synapses —particularly in aspiny shafts, which are likely to be originated from interneurons —, found in the present study in SLM compared to other CA1 layers is related to a particular synaptic circuit organization involving certain types of interneurons located in this layer remains to be elucidated.

## Materials and methods

### Key resources table

| Reagent type (species) or resource | Designation | Source or reference | Identifiers | Additional information |
|---|---|---|---|---|
| Chemical compound, drug | Paraformaldehyde | Sigma-Aldrich | Sigma-Aldrich: 24898648 | |
| Chemical compound, drug | Glutaraldehyde 25% EM | TAAB | TAAB: G002 | |
| Chemical compound, drug | Calcium chloride | Sigma-Aldrich | Sigma-Aldrich C2661 | |

*Continued on next page*

*Continued*

| Reagent type (species) or resource | Designation | Source or reference | Identifiers | Additional information |
|---|---|---|---|---|
| Chemical compound, drug | Sodium cacodylate trihydrate | Sigma-Aldrich | Sigma-Aldrich C0250 | |
| Chemical compound, drug | Osmium tetroxide | Sigma-Aldrich | Sigma-Aldrich O5500 | |
| Chemical compound, drug | Potassium ferricyanide | Probus | Probus: 23345 | |
| Chemical compound, drug | Uranyl acetate | EMS | EMS: 8473 | |
| Chemical compound, drug | Araldite | TAAB | TAAB: E201 | |
| Chemical compound, drug | Toluidine blue | Merck | Merck: 115930 | |
| Chemical compound, drug | Sodium borate | Panreac | Panreac: 141644 | |
| Chemical compound, drug | Silver paint | EMS | EMS: 12630 | |
| Software, algorithm | Stereo Investigator stereological package | MicroBrightField Inc | Version 8.0 | |
| Software, algorithm | Espina Interactive Neuron Analyzer | EspINA https://cajalbbp.es/espina | Version 2.4.1 | |
| Software, algorithm | ImageJ | ImageJ http://imagej.nih.gov/ij/ | ImageJ 1.51 | |
| Software, algorithm | GraphPad Prism | GraphPad Prism https://graphpad.com | Version 7.00 | |
| Software, algorithm | IBM SPSS Statistics for Windows | SPSS Software https://www.ibm.com/es-es/analytics/spss-statistics-software | Version 24.0 | |
| Software, algorithm | R project | R software http://www.R-project.org | Version 3.5.1 | |
| Software, algorithm | Easyfit Proffesional | Easyfit http://www.mathwave.com/es/home.html | Version 5.5 | |

## Sampling procedure

Human brain tissue was obtained from autopsies (with short post-mortem delays of less than 4.5 hours) from 5 subjects with no recorded neurological or psychiatric alterations (supplied by Unidad Asociada Neuromax, Laboratorio de Neuroanatomía Humana, Facultad de Medicina, Universidad de Castilla-La Mancha, Albacete and the Laboratorio Cajal de Circuitos Corticales UPM-CSIC, Madrid, Spain) (*Supplementary file 1AE*). The consent of the individuals was obtained and the sampling procedure was approved by the Institutional Ethical Committee of the Albacete University Hospital. The tissue was obtained following national laws and international ethical and technical guidelines on the

use of human samples for biomedical research purposes. Brain tissue was analyzed for Braak stage (**Braak and Braak, 1991**) and CERAD neuropathological diagnosis (**Mirra et al., 1991**) and assigned a zero score. Nevertheless, case AB1 showed sparse tau-immunoreactive cells in the hippocampal formation and case AB4 showed a relatively high number of amyloid plaques mainly located in the subicular and the parahippocampal regions. Tissue from some of these human brains has been used in previous unrelated studies (**Benavides-Piccione et al., 2020**; **Tapia-González et al., 2020**).

After extraction, brain tissue was fixed in cold 4% paraformaldehyde (Sigma-Aldrich, St Louis, MO, USA) in 0.1 M sodium phosphate buffer (PB; Panreac, 131965, Spain), pH 7.4, for 24 h. Subsequently, the block of tissue containing the hippocampus was washed in PB and coronal 150 µm-sections were obtained with a vibratome (Vibratome Sectioning System, VT1200S Vibratome, Leica Biosystems, Germany).

## Tissue processing for EM

Coronal sections from the hippocampal body (**Duvernoy, 2005**) containing the CA1 region were selected and postfixed for 48 h in a solution of 2% paraformaldehyde, 2.5% glutaraldehyde (TAAB, G002, UK) and 0.003% $CaCl_2$ (Sigma, C-2661-500G, Germany) in 0.1 M sodium cacodylate buffer (Sigma, C0250-500G, Germany). The sections were treated with 1% $OsO_4$ (Sigma, O5500, Germany), 0.1% ferrocyanide potassium (Probus, 23345, Spain) and 0.003% $CaCl_2$ in sodium cacodylate buffer (0.1 M) for 1h at room temperature. Sections were then stained with 1% uranyl acetate (EMS, 8473, USA), dehydrated, and flat embedded in Araldite (TAAB, E021, UK) for 48 h at 60°C. Embedded sections were glued onto a blank Araldite block and trimmed. Semithin sections (1 µm) were obtained from the surface of the block and stained with 1% toluidine blue (Merck, 115930, Germany) in 1% sodium borate (Panreac, 141644, Spain). The blocks containing the embedded tissue were then glued onto a sample stub using conductive adhesive tabs (EMS 77825-09, Hatfield, PA, USA). All the surfaces of the block —except for the one to be studied (the top surface)— were covered with silver paint (EMS 12630, Hatfield, PA, USA) to prevent charging artifacts. The stubs with the mounted blocks were then placed into a sputter coater (Emitech K575X, Quorum Emitech, Ashford, Kent, UK) and the top surface was coated with a 10–20 nm thick layer of gold/palladium to facilitate charge dissipation.

## Layer delimitation

The exact location of all CA1 layers was determined by examining 1% toluidine blue-stained semithin sections under a light microscope (**Figure 1**). More specifically, the medial portion of the CA1 region was analyzed. From its deepest level to the surface (i.e., from the ventricular cavity towards the vestigial hippocampal sulcus), the cornu ammonis may be divided into five layers: the alveus, *stratum oriens* (SO), *stratum pyramidale* (SP), *stratum radiatum* (SR) and *stratum lacunosum-moleculare* (SLM) (**Duvernoy, 2005**). Within the SP, two sublayers were defined by dividing the layer into a deeper part (dSP; close to the ventricular cavity) and a more superficial part (sSP; close to the vestigial hippocampal sulcus; **Figures 1a-b** and **7**; **Andrioli et al., 2007**; **Braak, 1974**).

To calculate the thickness of the layers, they were delimited using toluidine blue-stained semithin section adjacent to the block surface (**Figure 1**). Three measures per case were taken at different medio-lateral levels of CA1. This analysis was performed using ImageJ (ImageJ 1.51; NIH, USA; http://imagej.nih.gov/ij/).

## Volume fraction estimation of cortical elements

From each case, three semithin sections (1 µm thick; stained with 1% toluidine blue) were used to estimate the volume fraction occupied by blood vessels, cell bodies, and neuropil in each layer. This estimation was performed applying the Cavalieri principle (**Gundersen et al., 1988**) by point counting using the integrated Stereo Investigator stereological package (Version 8.0, MicroBrightField Inc, VT, USA) attached to an Olympus light microscope (Olympus, Bellerup, Denmark) at 40x magnification (**Figure 1—figure supplement 2a**). A grid whose points had an associated area of 400 µm$^2$ was overlaid over each semithin section to determine the $V_v$ occupied by different elements: blood vessels, glia, neurons and neuropil. $V_v$ occupied by the neuropil was estimated with the following formula: $V_v$ neuropil = 100 - ($V_v$ blood vessels + $V_v$ glia + $V_v$ neurons).

## FIB/SEM technology

A 3D EM study of the samples was conducted using combined FIB/SEM technology (Crossbeam 540 electron microscope, Carl Zeiss NTS GmbH, Oberkochen, Germany), as described in *Merchán-Pérez et al., 2009*; with some modifications. We used a 7-nA ion beam current with a 30-kV acceleration potential and a first coarse cross-section was milled as a viewing channel for SEM observation. The exposed surface of this cross-section was fine polished by lowering the ion beam current to 700 pA. Subsequently, layers from the fine polished cross-section were serially milled by scanning the ion beam parallel to the surface of the cutting plane using the same ion beam current. To mill each layer, the ion beam was automatically moved closer to the surface of the cross-section by preset increments of 20 nm, which corresponded to the thickness of the layers. After the removal of each slice, the milling process was paused and the freshly exposed surface was imaged using a 1.8-nA probe current with a 1.7-kV acceleration potential using the in-column energy-selective backscattered electron detector (EsB). The dwell time was 50 ns. The milling and imaging processes were continuously repeated and long series of images were acquired via a fully automated procedure. The quality and resolution of the images is similar to those achieved with TEM (*Merchán-Pérez et al., 2009*; *Figure 1c-d*). This study was conducted in the neuropil —that is, avoiding the neuronal and glial somata, blood vessels, large dendrites and myelinated axons— where most synaptic contacts take place (*DeFelipe et al., 1999*).

Image resolution in the xy plane was 5 nm/pixel. Resolution in the z-axis (section thickness) was 20 nm, and image size was 2,048 x 1,536 pixels. These parameters were optimized to make it possible to obtain a large enough field of view where the different types of synapses can be clearly identified in a reasonable amount of time (12 h per stack of images). The volume per stack ranged from 356 $\mu m^3$ to 727 $\mu m^3$ (225 and 459 images, respectively). All measurements were corrected for the tissue shrinkage that occurs during osmication and plastic-embedding of the vibratome sections containing the area of interest, as described by *Merchán-Pérez et al., 2009*. We measured the surface area and thickness of the vibratome sections with Stereo Investigator (MBF Bioscience, Williston, VT, USA), both before and after they were processed for EM (*Oorschot et al., 1991*). The surface area after processing was divided by the value before processing to obtain an area shrinkage factor ($p^2$) of 0.933. The linear shrinkage factor for measurements in the plane of the section (p) was therefore 0.966. The shrinkage factor in the z-axis was 0.901. In addition, the total volume was corrected for the presence of fixation artifacts, which did not affect the accurate identification and quantitation of synapses (i.e., swollen neuronal or glial processes). The volume occupied by these artifacts was calculated applying the Cavalieri principle (*Gundersen et al., 1988*) and was discounted from the volume of the stacks of images to avoid underestimation of the number of synapses per volume. Specifically, a stereological grid with an associated area per point of 400,000 $nm^2$ was superimposed onto each FIB/SEM stack with the Image J Stereology Toolset (*Mironov, 2017*). Estimations were made every 20th section of each stack. Volume fraction estimation was performed by point counting using the Cavalieri principle (*Gundersen et al., 1988*), in a similar fashion to the volume fraction estimation of cortical elements in 1% toluidine blue-stained semithin sections outlined above (see 'Volume fraction estimation of cortical elements'). A fixation artifact factor was calculated for each FIB/SEM stack (ranging from 0 to 20% of the stack volume) and was applied to each individual FIB/SEM stack. All parameters measured were corrected to obtain an estimate of the pre-processing values. The shrinkage factor was used to correct the synaptic apposition surface (SAS) area and perimeter data, while both the shrinkage and the fixation artifact factors were used to correct synaptic density values. Corrected and uncorrected data for each parameter are shown in *Table 1*.

A total of 75 stacks of images from all layers of the CA1 field were obtained (3 stacks per case and region in the 5 cases, with a total volume studied of 29,322 $\mu m^3$) (*Figure 1b*).

## 3D analysis of synapses
### Classification of synapses and postsynaptic target identification
EspINA software was used for the 3D segmentation and classification of synapses in the 75 stacks of images (*Espina Interactive Neuron Analyzer*, 2.4.1; Madrid, Spain; https://cajalbbp.es/espina/; *Morales et al., 2013*). As previously discussed in *Ascoli et al., 2008*, there is a consensus for classifying cortical synapses into AS (or type I) and SS (or type II) synapses. The main characteristic distinguishing these synapses is the prominent or thin post-synaptic density, respectively. Nevertheless, in

single sections, the synaptic cleft and the pre- and post-synaptic densities are often blurred if the plane of the section does not pass at right angles to the synaptic junction. Since the software EspINA allows navigation through the stack of images, it was possible to unambiguously identify every synapse as AS or SS based on the thickness of the PSD. Synapses with prominent PSDs are classified as AS, while thin PSDs are classified as SS (*Figure 1c-f*, *3* and *4*).

Additionally, based on the postsynaptic targets, synapses were further classified as axospinous synapses (synapses on dendritic spines) and axodendritic synapses (synapses on dendritic shafts). In the case of axospinous synapses, they were further subdivided into axospinous synapses on the head or on the neck of the spine. For axodendritic synapses, dendritic shafts were further classified as spiny (when dendritic spines could be observed emerging from the shaft) or aspiny. Only clearly identifiable postsynaptic elements were quantified (i.e., elements that were unambiguously identified from navigating through the stack of images; *Figure 4a-k*).

Finally, synapses were classified —according to the shape of their synaptic junction— into four categories, as described elsewhere (*Santuy et al., 2018b*). In short, synapses with a flat, disk-shaped PSD were classified as macular. A second category was established by the presence of an indentation in the perimeter (horseshoe-shaped synapses). Synaptic junctions with one or more holes in the PSD were referred to as perforated. Synaptic junctions with two or more physically discontinuous PSDs were categorized as fragmented (*Figure 2d*).

## Morphological and spatial measurements

The 3D segmentation of synaptic junctions includes both the presynaptic density (active zone; AZ) and the PSD. Since the AZ and the PSD are located face to face, their surface areas are very similar (correlation coefficients over 0.97; *Schikorski and Stevens, 1997*; *Schikorski and Stevens, 1999*). Thus, as previously described in *Morales et al., 2013*), they can be simplified to a single surface and represented as the surface of apposition between the AZ and the PSD. This surface can be extracted from the 3D segmented synaptic junction (*Morales et al., 2013*). For the sake of clarity, we have referred to this surface as the synaptic apposition surface (SAS). We consider SAS morphological measurements to be a better approach to the assessment of synaptic size than measurements obtained from the 3D segmented synaptic junctions (see *Morales et al., 2013* for more detailed information about SAS extraction and its relation to presynaptic density and PSD). We observed in our samples that the SAS area is highly correlated to the surface ($R^2$ = 0.96 for AS; $R^2$ = 0.97 for SS) and the volume ($R^2$ = 0.91 for AS; $R^2$ = 0.90 for SS) of the 3D segmented synaptic junctions.

The SAS area and perimeter of each synaptic junction was extracted with EspINA software to study morphological parameters regarding synapses. EspINA software also permits the quantitation of the curvature of the synapses as it adapts to the curvature of the synaptic junction. Specifically, curvature measurements are calculated as 1 minus the ratio between the projected area of the SAS and the area of the SAS (*Morales et al., 2011*). This measurement would be 0 in a flat SAS and would increase its value to a maximum of 1 as the SAS curvature increases.

The spatial distribution of synapses was determined by performing a Spatial Point Pattern analysis (*Anton-Sanchez et al., 2014*; *Merchán-Pérez et al., 2014*). The position of centroids of the synapses was compared to the Complete Spatial Randomness (CSR) model, which defines a situation where a point is equally probable to occur at any location within a given volume. For each stack of images, functions F, G and K were calculated (*Blazquez-Llorca et al., 2015*). In addition, the distance of every synapse to its nearest synapse was measured. This study was carried out using Spatstat package and R Project software (*Baddeley et al., 2015*).

## Statistical analysis

Statistical analysis of the data was carried out using GraphPad Prism statistical package (Prism 7.00 for Windows, GraphPad Software Inc, USA), SPSS software (IBM SPSS Statistics for Windows, Version 24.0. Armonk, NY: IBM Corp), Easyfit Proffesional 5.5 (MathWave Technologies) and R Project software (R 3.5.1; Bell Laboratories, NJ, USA; http://www.R-project.org). Differences in the $V_v$ occupied by cortical elements; synaptic density; and morphological and spatial parameters were analyzed performing either a two-sided, one-way analysis of variance (ANOVA), with Tukey post hoc corrections, or Mann-Whitney U (MW) nonparametric test, as appropriate. Frequency distributions were analyzed using Kolmogorov-Smirnov (KS) nonparametric tests. Chi-squared ($\chi^2$) tests were used for

contingency tables. In general, for any contingency table, the expected frequency for a cell in the $i^{th}$ row and the $j^{th}$ column is $E_{ij} = T_i T_j / T$, where $T_i$ is the marginal total for the $i^{th}$ row, $T_j$ is the marginal total for the $j^{th}$ column, and T is the total number of observations. $\chi^2$ tests of association were applied to these tables (*Sharpe, 2015*). The criterion for statistical significance was considered to be met for $p<0.05$ when the sample size was equal to the number of subjects (i.e., ANOVA and MW tests), and for $p<0.001$ when the sample size was equal to the number of synapses (i.e., KS and $\chi^2$ tests), in order to avoid overestimation of the differences due to a very big sample size.

## Acknowledgements

We would like to thank Carmen Álvarez, Miriam Martín and Lorena Valdés for their helpful technical assistance and Nick Guthrie for his excellent text editing. This work was supported by grants from the following entities: Centro de Investigación en Red sobre Enfermedades Neurodegenerativas (CIBERNED, CB06/05/0066, Spain); the Spanish 'Ministerio de Ciencia, Innovación y Universidades' (grant PGC2018-094307-B-I00 and the Cajal Blue Brain Project [the Spanish partner of the Blue Brain Project initiative from EPFL, Switzerland]); the European Union's Horizon 2020 Research and Innovation Programme under grant agreement No. 785907 (Human Brain Project, SGA2), the Alzheimer's Association (ZEN-15–321663) and the UNED (Plan de Promoción de la Investigación, 2014–040-UNED-POST). MM-C was awarded a research fellowship from the Spanish Ministry of Education, Culture and Sports (contract FPU14/02245).

## Additional information

### Funding

| Funder | Grant reference number | Author |
|---|---|---|
| Centro de Investigación Biomédica en Red sobre Enfermedades Neurodegenerativas | CB06/05/0066 | Javier DeFelipe |
| Spanish Ministry of Science and Innovation | PGC2018-094307-B-I00 | Javier DeFelipe |
| Spanish Ministry of Science and Innovation | Cajal Blue Brain Project | Javier DeFelipe |
| Horizon 2020 Framework Programme | 785907 | Javier DeFelipe |
| Alzheimer's Association | ZEN-15-321663 | Javier DeFelipe |
| Universidad Nacional de Educación a Distancia | 2014-040-UNED-POST | Lidia Blazquez-Llorca |
| Ministry of Education, Culture and Sports, Spain | FPU14/02245 | Marta Montero-Crespo |

The funders had no role in study design, data collection and interpretation, or the decision to submit the work for publication.

### Author contributions

Marta Montero-Crespo, Conceptualization, Data curation, Software, Formal analysis, Investigation, Methodology, Writing - original draft, Writing - review and editing; Marta Dominguez-Alvaro, Patricia Rondon-Carrillo, Data curation, Software; Lidia Alonso-Nanclares, Conceptualization, Formal analysis, Supervision, Validation, Methodology; Javier DeFelipe, Conceptualization, Resources, Supervision, Funding acquisition, Validation, Methodology, Project administration, Writing - review and editing; Lidia Blazquez-Llorca, Conceptualization, Data curation, Formal analysis, Supervision, Validation, Investigation, Methodology, Writing - original draft, Writing - review and editing

Author ORCIDs
Marta Montero-Crespo https://orcid.org/0000-0002-7895-5961
Marta Dominguez-Alvaro https://orcid.org/0000-0002-4057-5971
Lidia Alonso-Nanclares http://orcid.org/0000-0003-2649-7097
Javier DeFelipe https://orcid.org/0000-0001-5484-0660
Lidia Blazquez-Llorca https://orcid.org/0000-0002-4865-8974

## Ethics

Human subjects: The samples were obtained under informed consent and following the guidelines of the ethics committee of the institutions involved. The consent of the individuals was obtained and the sampling procedure was approved by the Institutional Ethical Committee of the Albacete University Hospital. The tissue was obtained following national laws and international ethical and technical guidelines on the use of human samples for biomedical research purposes. Tissue from some of these human brains has been used in previous unrelated studies (Benavides-Piccione et al., 2020; Tapia-González et al., 2020).

## Decision letter and Author response

Decision letter https://doi.org/10.7554/eLife.57013.sa1
Author response https://doi.org/10.7554/eLife.57013.sa2

# Additional files

## Supplementary files
- Supplementary file 1. Supplementary tables.
- Transparent reporting form

## Data availability

Most data is available in the main text or the supplementary materials. The datasets used and analyzed during the current study are published on the EBRAINS Knowledge Graph (DOI: https://doi.org/10.25493/6HRE-F2Y and https://doi.org/10.25493/NRFB-7N5).

The following datasets were generated:

| Author(s) | Year | Dataset title | Dataset URL | Database and Identifier |
|---|---|---|---|---|
| Domínguez-Álvaro M, Montero-Crespo M, Alonso-Nanclares L, Rodriguez R, DeFelipe J | 2020 | Densities and 3D distributions of synapses using FIB/SEM imaging in the human Hippocampus (CA1) - Extension with additional subregions | https://doi.org/10.25493/NRFB-7N5 | EBRAINS , 10.25493/NRFB-7N5 |
| Dominguez-Alvaro M, Montero-Crespo M, Alonso-Nanclares L, Rodriguez R, DeFelipe J | 2020 | Densities and 3D distributions of synapses using FIB/SEM imaging in the human hippocampus (CA1) | https://doi.org/10.25493/6HRE-F2Y | EBRAINS, 10.25493/6HRE-F2Y |

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

## Appendix 1

### Notes on human CA1 connectivity

The major input to CA1 originates in the EC (glutamatergic) (a−b in *Figure 7*). Neurons located in layer III of the EC project directly to SLM (a in *Figure 7*; *Witter and Amaral, 1991* [1 in *Figure 7*]), while layer II neurons project indirectly to SO and SR, but, unlike in rodents, they also project to SP (b in *Figure 7*), via DG and CA3 Schaffer collaterals (SC) (glutamatergic) (*Kondo et al., 2009* [2 in *Figure 7*]). Additionally, minor projections —from the amygdala (glutamatergic), the VTA (dopaminergic), the locus coeruleus (noradrenergic), the raphe nuclei (serotonergic), the medial septal nucleus (Ch1) (cholinergic), the vertical limb of the diagonal band of Broca (cholinergic) and the laterodorsal tegmental nucleus (cholinergic)— have been described to arrive in all layers of CA1 (c−f in *Figure 7*). However, some of these projections (amygdala, raphe nuclei and laterodorsal tegmental nucleus) have been described as being especially numerous in the SLM (c in *Figure 7*, represented with a '+++") (*Aggleton, 1986* [3 in *Figure 7*]; *Alonso and Amaral, 1995* [4 in *Figure 7*]; *Amaral and Cowan, 1980* [5 in *Figure 7*]; *Barone et al., 1994* [6 in *Figure 7*]; *De Lacalle et al., 1994* [7 in *Figure 7*]; *Del Fiacco et al., 1987* [8 in *Figure 7*]; *Green and Mesulam, 1988* [9 in *Figure 7*]; *Ihara et al., 1988* [10 in *Figure 7*]; *Iritani et al., 1989* [11 in *Figure 7*]; *Klimek et al., 1999* [12 in *Figure 7*]; *Mesulam et al., 1983* [13 in *Figure 7*]; *Powers et al., 1988* [14 in *Figure 7*]; *Samson et al., 1990* [15 in *Figure 7*]; *Wang and Barbas, 2018* [16 in *Figure 7*]; *Wilson and Molliver, 1991* [17 in *Figure 7*]). In addition, differences have been reported regarding the density of immunoreactive ChAT fibers between human and monkey (higher density of fiber labeling in SP and SR in human CA1 and higher density in SLM and SO in monkey CA1) (*Alonso and Amaral, 1995* [4 in *Figure 7*]; *De Lacalle et al., 1994* [7 in *Figure 7*]). Minor direct projections have also been reported from the medial septal nuclei (GABAergic) to SO (mainly innervating interneurons) (f in *Figure 7*; *Gulyás et al., 1991* [18 in *Figure 7*]). In addition, minor projections have also been reported from several cortical regions (glutamatergic) including: the medial temporal cortex (TH, TF1, TF2 — posterior parahippocampal areas; 35, 36 — perirhinal cortex; TEav, TEpv —ventral inferotemporal areas) and the parietal cortex (7a and 7b) mainly to SLM (c in *Figure 7*; *Rockland and Van Hoesen, 1999* [19 in *Figure 7*]; *Yukie, 2000* [20 in *Figure 7*]) and from the prefrontal cortex mainly to SO and SP (e,f in *Figure 7*; *Leichnetz and Astruc, 1975* [21 in *Figure 7*]). Other regions have been observed to project to the hippocampal formation of the monkey but there is no specific information regarding possible direct projections to CA1 (for example, the claustrum; the substantia innominata; the basal nucleus of Meynert; the thalamus —specifically the anterior nuclear complex, the laterodorsal nucleus, the paraventricular and parataenial nuclei, the nucleus reuniens, and the nucleus centralis medialis—; the lateral preoptic and lateral hypothalamic areas; the supramammillary and retromammillary regions; the tegmental reticular fields; the nucleus reticularis tegmenti pontis; and the central gray — for more details, see *Amaral and Cowan, 1980* [4 in *Figure 7*]). For example, the midline thalamic nuclei (mostly Reuniens nucleus) have been observed to project mainly to the SLM in the rat CA1 but there is no information regarding primates (see *Insausti and Amaral, 2012*).

Regarding the longitudinal axis of the hippocampus (bottom of the diagram in *Figure 7*; modified from *Strange et al., 2014*), there are some examples in which it has been described that primate CA1 receives different inputs along this axis. For example, (*i*) lateral portions of the EC project to caudal levels of the recipient fields and more medial parts of the EC project to progressively more rostral portions (*Witter and Amaral, 1991* [1 in *Figure 7*]); (*ii*) CA3 projections to CA1 extend very widely both rostrally and caudally. However, the projections from CA3 neurons located at the level of the uncus are restricted to rostral CA1, whereas neurons in proximal CA3 project to caudal CA1 and neurons in distal CA3 project to rostral CA1 (*Kondo et al., 2009* [2 in *Figure 7*]); (*iii*) the accessory and lateral basal nucleus of the amygdala project to rostral CA1, while the medial basal nucleus projects along the whole longitudinal axis (*Aggleton, 1986* [3 in *Figure 7*]); (*iv*) the temporal TH cortex projects to rostral CA1, the temporal TF2 cortex to medial CA1 and temporal TF1, TF2, 35,

36, TEav and TEpv and the parietal (7a and 7b) cortices project to caudal CA1 (*Rockland and Van Hoesen, 1999* [19 in *Figure 7*]; *Yukie, 2000* [20 in *Figure 7*]); (v) the cholinergic innervation from medial septal nuclei and diagonal band of Broca is mainly present in rostral CA1 (*Alonso and Amaral, 1995* [4 in *Figure 7*]; see also *Aggleton, 2012*).

The main output of the CA1 region is the subiculum (i in *Figure 7*; *Insausti and Amaral, 2012*). However, CA1 has also been reported to project directly to other brain areas (j in *Figure 7*), such as the amygdala, the orbitofrontal cortex (areas 11 and 13), the medial prefrontal cortex (areas 14, 25 and 32), the medial temporal area TE and the temporal pole TG, to a similar degree as the subicular projection (*Aggleton, 1986*; *Aggleton, 2012*; *Barbas and Blatt, 1995*; *Carmichael and Price, 1995*; *Cavada et al., 2000*; *Insausti and Muñoz, 2001*; *Iwai and Yukie, 1988*; *Morecraft et al., 1992*; *Rosene and Van Hoesen, 1977*). Furthermore, CA1 projects to other areas including layer V of both the EC and the perirhinal cortex, as well as layers III and V of the parahippocampal cortex, where CA1 becomes the major hippocampal source of projections (*Aggleton, 2012*; *Blatt and Rosene, 1998*; *Insausti and Muñoz, 2001*; *Saunders and Rosene, 1988*; *Yukie, 2000*). Additionally, CA1 sends projections to the medial septal nucleus (Ch1), vertical limb of the diagonal band of Broca (Ch2) and the nucleus accumbens, and it is known that these projections are not shared with the subiculum (*Aggleton, 2012*; *Aggleton et al., 1987*; *Friedman et al., 2002*; *Rosene and Van Hoesen, 1977*). According to *Aggleton et al., 1986*, CA1 does not project to the thalamus.

Finally, there are also differences in the hippocampal projections along the longitudinal hippocampal axis (*Aggleton, 2012*; *Leonardo et al., 2006*; *Strange et al., 2014*), For example, more rostral parts of CA1 project mainly to rostral and medial parts of the perirhinal cortex, as well as to the lateral septum, the amygdala, nucleus accumbens, the orbitofrontal cortex and the medial prefrontal cortex (*Aggleton, 1986*; *Aggleton, 2012*; *Barbas and Blatt, 1995*; *Blatt and Rosene, 1998*; *Carmichael and Price, 1995*; *Friedman et al., 2002*; *Saunders and Rosene, 1988*). Mid portions of CA1 project to the EC, while the most posterior parts of CA1 send projections mainly to the caudal and lateral portions of the parahippocampal cortex and, to a lesser extent, to dorsal and medial parts of the septum (*Aggleton, 2012*; *Aggleton et al., 2005*; *Insausti and Muñoz, 2001*; *Yukie, 2000*).

