## [Decision Letter]

**Acceptance summary:**

Your manuscript presents the most extensive description of the human CA1 neuropil to date, and thus will serve as important reference for future human and comparative studies.

**Decision letter after peer review:**

Thank you for submitting your article "Three-dimensional synaptic organization of the human hippocampal CA1 field" for consideration by *eLife*. Your article has been reviewed by two peer reviewers, and the evaluation has been overseen by a Reviewing Editor and Laura Colgin as the Senior Editor. The reviewers have opted to remain anonymous.

The reviewers have discussed the reviews with one another and the Reviewing Editor has drafted this decision to help you prepare a revised submission.

Your manuscript quantifies synapse variability across the layers of the human hippocampal CA1 field collected from human brain autopsies. The reviewers agree that your work present the most extensive description of the human CA1 neuropil to date, and thus will serve as important reference for future human and comparative studies.

While the reviewers are excited about your work, they also found that minor revisions to address clarity with additional analyses would benefit the manuscript.

Reviewer #1:

It is believed that a thorough understanding of how neural circuit works depends on the dissection of its synaptic-level organization. Electron microscopy (EM) remains the golden standard in mapping synaptic connections, but reconstructing synaptic circuits from even a modest volume of brain tissue remains technically challenging. Montero-Crespo et al. used the novel focused ion beam/scanning electron microscopy (FIB/SEM) to reconstruct a large number (24,752) of synapses in the hippocampal CA1 region from human brain autopsy specimens. They analyzed a variety of structural parameters including synaptic density, spatial distribution, synapse type, shape, size, and postsynaptic targets. They found remarkable differences in these parameters across CA1 layers.

This study provides a wealth of data on synaptic connections in hippocampal CA1 region, which will be highly informative for future studies on its physiological and computational roles. Moreover, this paper examines brain autopsy samples from deceased humans without recorded neurological or psychiatric alterations. Given the scarcity of such human brain tissues available for this type of study, results from this paper are very valuable.

However, a lot of interesting questions could have been addressed by the author using the data they obtained. In particular, as FIB/SEM intrinsically provides perfect alignment between consecutive Z sections, it would be really nice to leverage this feature and trace some dendritic and axonal segments and to address questions that relate the features of individual synapses to their parent neuronal processes.

1) What is the spatial distribution of excitatory and inhibitory synapses along a dendritic segment? Do they appear to follow an independent, uniform distribution, or is there any sign of clustering, either between synapses in the same category, or between excitatory and inhibitory synapses? Does the spatial distribution differ between layers?

2) What is the relationship between the synaptic apposition surface (SAS) and the size of PSD? Are the total number of synapses (excitatory or inhibitory) and summed synaptic areas (each type of synapses) per unit length of dendritic segment approximately constant, as suggested by Bromer et al., 2018? If so, is the constant universal across layers or specific to each layer?

3) What is the density of "potential synapses", i.e., sites where an axon comes within a sufficiently short distance from the dendritic shaft to make synaptic connectivity attainable, as defined in Stepanyants et al. (Neuron 34(2): 275-88, 2002)? Does the "filling fraction" (ibid.) agree or differ from the value obtained from mouse hippocampal connectomic reconstruction in Mishchenko et al. (Neuron 67(6): 1009-1020, 2010)?

4) If SAS or PSD size is used as a proxy for synaptic strength, are the author's data consistent with a "quantized" distribution of synaptic strength, as suggested by Bartol et al., 2015 and Bromer et al., 2018? Do excitatory and inhibitory synapses follow the same or different distributions? Do synapses in different layers follow the same or different distributions?

Reviewer #2:

Montero-Crespo et al. quantify synapse variability across the layers of the human hippocampal CA1 field collected from 5 human brain autopsies. This is clearly the most extensive description of the human CA1 neuropil to date. While primarily being solely descriptive in nature, I think it is nevertheless a benchmark paper that will serve as a reference for future human and comparative studies. The major findings include differences in synaptic density across CA1 layers, differences in synapse shape between layers and specializations particular to the SLM layer. In the Discussion, the authors do a reasonable job trying to relate their findings to connectivity information from primate studies. If anything, I would recommend expanding the discussion comparing the results to those in rodents since I think that would broaden the interest. I think perhaps adding a table to a main figure summarizing the most important species differences would also help.

I had just a few questions/points:

1) Please add the imaging parameters of the FIB/SEM (e.g., electron energy and current, dwell time, ion beam current, etc…)

2) I didn't really understand the correction applied to compensate for fixation artifacts (subsection “FIB/SEM technology”). Please expand. It sounds like two corrections were performed, one for EM/embedding shrinkage of around 7%, and then a stack specific correction for fixation artifacts. I didn't understand how the fixation artifact was measured and what was actually done to the FIB/SEM volumes to correct for it. Since these corrections are critical to the interpretation of the data (such as synapses densities, etc…), an explicit example of how the corrections were done is important to show.

3) Were all judgments about synapse type and shape made by a single annotator or as a consensus among multiple annotators?

4) Did the authors ever find multiple synapses onto a postsynaptic target originating from the same presynaptic axon? Or were the volumes too small for this?

5) Typo in the last paragraph of the subsection “Relation between synaptic inputs and synaptic organization of each layer”, should be 'afferents' I think.

---

## [Author Response]

Reviewer #1:[…] This study provides a wealth of data on synaptic connections in hippocampal CA1 region, which will be highly informative for future studies on its physiological and computational roles. Moreover, this paper examines brain autopsy samples from deceased humans without recorded neurological or psychiatric alterations. Given the scarcity of such human brain tissues available for this type of study, results from this paper are very valuable.However, a lot of interesting questions could have been addressed by the author using the data they obtained. In particular, as FIB/SEM intrinsically provides perfect alignment between consecutive Z sections, it would be really nice to leverage this feature and trace some dendritic and axonal segments and to address questions that relate the features of individual synapses to their parent neuronal processes.

We would like to thank the reviewer for his/her positive comments, especially considering it took 2 people 3 years of full-time work to obtain the data provided in the present article. We completely agree with the reviewer that tracing the dendritic and axonal segments to get further information about individual segmented synapses is of great importance in terms of the human connectome. However, performing this study with thousands of synapses, dendrites and axons would be very time-consuming and technically challenging. Certainly, we could carry out this study with a relatively small subset of the reconstructed synapses to accelerate the process, but the obtained connectivity data would be less robust than if we were to use all of the reconstructed synapses that we were able to achieve. For these reasons, our laboratory is currently developing a software tool that will allow us to accurately and reliably trace and reconstruct a large number of axonal and dendritic segments contained within the FIB/SEM stacks. In our opinion, this analysis would make it possible to accurately answer important questions regarding pre- and post-synaptic elements in detail and in relation to their parental neuronal processes. Thus, we are expecting to answer these questions in the near future. Respectfully, we would like to leave the answer to questions 1 to 3 (from reviewer 1) and question 4 (from reviewer 2) for a future study.

1) What is the spatial distribution of excitatory and inhibitory synapses along a dendritic segment? Do they appear to follow an independent, uniform distribution, or is there any sign of clustering, either between synapses in the same category, or between excitatory and inhibitory synapses? Does the spatial distribution differ between layers?2) What is the relationship between the synaptic apposition surface (SAS) and the size of PSD? Are the total number of synapses (excitatory or inhibitory) and summed synaptic areas (each type of synapses) per unit length of dendritic segment approximately constant, as suggested by Bromer et al., 2018? If so, is the constant universal across layers or specific to each layer?

Regarding the relationship between the synaptic apposition surface (SAS) and the size of the PSD (Materials and methods), we have added the following paragraph to make some points regarding the SAS clearer:

“The 3D segmentation of synaptic junctions includes both the presynaptic density (active zone; AZ) and the PSD. […] We observed in our samples that the SAS area is highly correlated to the surface (R^2^ = 0.96 for AS; R^2^ = 0.97 for SS) and the volume (R^2^ = 0.91 for AS; R^2^ = 0.90 for SS) of the 3D segmented synaptic junctions.”

3) What is the density of "potential synapses", i.e., sites where an axon comes within a sufficiently short distance from the dendritic shaft to make synaptic connectivity attainable, as defined in Stepanyants et al. (Neuron 34(2): 275-88, 2002)? Does the "filling fraction" (ibid.) agree or differ from the value obtained from mouse hippocampal connectomic reconstruction in Mishchenko et al. (Neuron 67(6): 1009-1020, 2010)?4) If SAS or PSD size is used as a proxy for synaptic strength, are the author's data consistent with a "quantized" distribution of synaptic strength, as suggested by Bartol et al., 2015 and Bromer et al., 2018? Do excitatory and inhibitory synapses follow the same or different distributions? Do synapses in different layers follow the same or different distributions?

Thank you for this comment. We have now added further information to explain that excitatory and inhibitory synapses follow the same distribution and there are not differences between layers.

We have added the following paragraph to the Results:

“Furthermore, we found that both types of synapses (AS and SS) fit log-normal or log-logistic probability density functions. These distributions, with some variations in the parameters of the functions (Supplementary file 1L), were found in each layer and the whole CA1 (all layers pooled together) for both AS and SS (Supplementary file 1L; Figure 6—figure supplement 2).”

We added the following paragraph to the Discussion:

“The size of both types of synaptic junctions (AS and SS) fit log-normal or loglogistic probability density functions (see Figure 6—figure supplement 2). […] Specifically, activity-dependent processes seem to primarily dictate the scale rather than the shape of synaptic size distributions (Hazan and Ziv, 2020).”

In the Materials and methods, the sentence:

“Statistical analysis of the data was carried out using GraphPad Prism statistical package (Prism 7.00 for Windows, GraphPad Software Inc, USA), SPSS software (IBM SPSS Statistics for Windows, Version 24.0. Armonk, NY: IBM Corp), and R Project software (R 3.5.1; Bell Laboratories, NJ, USA; http://www.R-project.org).”

has been changed to

“Statistical analysis of the data was carried out using GraphPad Prism statistical package (Prism 7.00 for Windows, GraphPad Software Inc, USA), SPSS software (IBM SPSS Statistics for Windows, Version 24.0. Armonk, NY: IBM Corp), Easyfit Professional 5.5 (MathWave Technologies) and R Project software (R 3.5.1; Bell Laboratories, NJ, USA; http://www.R-project.org).”

We have included a new figure (Figure 6—figure supplement 2).

Furthermore, some changes have been made in Supplementary file 1L to include the parameters α, β and γ of the log-logistic 3P curves and the parameters σ and μ of the log-normal curves.

Reviewer #2:Montero-Crespo et al. quantify synapse variability across the layers of the human hippocampal CA1 field collected from 5 human brain autopsies. This is clearly the most extensive description of the human CA1 neuropil to date. While primarily being solely descriptive in nature, I think it is nevertheless a benchmark paper that will serve as a reference for future human and comparative studies. The major findings include differences in synaptic density across CA1 layers, differences in synapse shape between layers and specializations particular to the SLM layer. In the Discussion, the authors do a reasonable job trying to relate their findings to connectivity information from primate studies. If anything, I would recommend expanding the discussion comparing the results to those in rodents since I think that would broaden the interest. I think perhaps adding a table to a main figure summarizing the most important species differences would also help.

We would like to thank the reviewer for his/her support. Certainly, one of our main interests is in the comparative studies. Indeed, we are planning to write a review paper dealing with comparative analysis of the synaptic organization in the CA1 neuropil of humans compared to CA1 in rats and mice. Importantly, estimations of synaptic density, size and types of synapses, using conventional electron microscopy (TEM) – which is the technique used in most EM studies – is based on the analysis of relatively few serial EM sections or stereological approaches, whereas with FIB/SEM it is possible to obtain multiple samples with thousands of serial sections with the number of synapses being directly counted in the stacks of images. Thus, at present we are performing comparative studies between species and techniques in order to get a clearer picture of these studies. For example, we are comparing the synaptic organization in the CA1 of rats and mice using FIB/SEM examined in our lab with previous data using TEM in the same species. In addition, we are comparing the data obtained in the present study of the human CA1 with that obtained in the rodents. We hope to finish this review in the next few months.

I had just a few questions/points:1) Please add the imaging parameters of the FIB/SEM (e.g., electron energy and current, dwell time, ion beam current, etc…)

We have added additional information regarding the sample preparation for FIB/SEM and FIB/SEM imaging.

The following information has been added:

“The blocks containing the embedded tissue were then glued onto a sample stub using conductive adhesive tabs (EMS 77825-09, Hatfield, PA, USA). […] The stubs with the mounted blocks were then placed into a sputter coater (Emitech K575X, Quorum Emitech, Ashford, Kent, UK) and the top surface was coated with a 10–20 nm thick layer of gold/palladium to facilitate charge dissipation.”

The following paragraph:

“A 3D EM study of the samples was conducted using combined FIB/SEM technology (Crossbeam 540 electron microscope, Carl Zeiss NTS GmbH, Oberkochen, Germany). […] This study was conducted in the neuropil – i.e., avoiding the neuronal and glial somata, blood vessels, large dendrites and myelinated axons – where most synaptic contacts take place (DeFelipe et al., 1999).”

has been changed to

“A 3D EM study of the samples was conducted using combined FIB/SEM technology (Crossbeam 540 electron microscope, Carl Zeiss NTS GmbH, Oberkochen, Germany), as described in (Merchán-Pérez et al., 2009) with some modifications. […] This study was conducted in the neuropil – i.e., avoiding the neuronal and glial somata, blood vessels, large dendrites and myelinated axons – where most synaptic contacts take place (DeFelipe et al., 1999).”

2) I didn't really understand the correction applied to compensate for fixation artifacts (subsection “FIB/SEM technology”). Please expand. It sounds like two corrections were performed, one for EM/embedding shrinkage of around 7%, and then a stack specific correction for fixation artifacts. I didn't understand how the fixation artifact was measured and what was actually done to the FIB/SEM volumes to correct for it. Since these corrections are critical to the interpretation of the data (such as synapses densities, etc…), an explicit example of how the corrections were done is important to show.

To make the two volume corrections applied clearer, the following paragraph:

“Volume measurements were corrected for tissue shrinkage due to EM processing (Merchán-Pérez et al., 2009) with a shrinkage factor obtained by dividing the area of the tissue after processing by the area of the same tissue before processing (p=0.933). […] Every FIB/SEM stack was examined, and the volume artifact ranged from 0 to 20% of the stack volume.”

has been changed to

“All measurements were corrected for the tissue shrinkage that occurs during osmication and plastic-embedding of the vibratome sections containing the area of interest, as described by (Merchán-Pérez et al., 2009). […] Corrected and uncorrected data for each parameter are shown in Table 1.”

3) Were all judgments about synapse type and shape made by a single annotator or as a consensus among multiple annotators?

The judgments about synapse type and shape were made initially by consensus among multiple annotators (3 annotators, MM-C, LB-L and MD-A). Since the judgment between annotators was highly uniform, synapses in the subsequent stacks (approximately half of the total stacks) were sorted by a single annotator (MM-C).

4) Did the authors ever find multiple synapses onto a postsynaptic target originating from the same presynaptic axon? Or were the volumes too small for this?

Please see our response to the first comment made by reviewer 1.

5) Typo in the last paragraph of the subsection “Relation between synaptic inputs and synaptic organization of each layer”, should be 'afferents' I think.

Thank you for pointing this out; the mistake has been corrected.